UPDATE ARTICLE

# Collective peroxide detoxification determines microbial mutation rate plasticity in *E. coli*

Rowan Green [1], Hejie Wang[2], Carol Botchey[2], Siu Nam Nancy Zhang[2], Charles Wadsworth[2], Francesca Tyrrell[2], James Letton[2], Andrew J. McBain [3], Pawel Paszek [2,4], Rok Krašovec [2]*, Christopher G. Knight [1]*

1 School of Natural Sciences, Faculty of Science & Engineering, University of Manchester, United Kingdom, 2 School of Biological Sciences, Faculty of Biology, Medicine & Health, University of Manchester, United Kingdom, 3 School of Health Sciences, Faculty of Biology Medicine & Health, University of Manchester, United Kingdom, 4 Department of Biosystems and Soft Matter, Institute of Fundamental Technological Research, Polish Academy of Sciences, Warsaw, Poland

* rok.krasovec@manchester.ac.uk (RK); chris.knight@manchester.ac.uk (CGK)

The Editors encourage authors to publish research updates to this article type. Please follow the link in the citation below to view any related articles.

## Abstract

Mutagenesis is responsive to many environmental factors. Evolution therefore depends on the environment not only for selection but also in determining the variation available in a population. One such environmental dependency is the inverse relationship between mutation rates and population density in many microbial species. Here, we determine the mechanism responsible for this mutation rate plasticity. Using dynamical computational modelling and in culture mutation rate estimation, we show that the negative relationship between mutation rate and population density arises from the collective ability of microbial populations to control concentrations of hydrogen peroxide. We demonstrate a loss of this density-associated mutation rate plasticity (DAMP) when *Escherichia coli* populations are deficient in the degradation of hydrogen peroxide. We further show that the reduction in mutation rate in denser populations is restored in peroxide degradation-deficient cells by the presence of wild-type cells in a mixed population. Together, these model-guided experiments provide a mechanistic explanation for DAMP, applicable across all domains of life, and frames mutation rate as a dynamic trait shaped by microbial community composition.

## Introduction

Uncovering the mechanisms behind environmentally responsive mutagenesis informs our understanding of evolution, notably antimicrobial resistance, where mutation supply can be critical [1,2]. Microbial mutation rates are responsive to a wide variety of environmental factors including population density [3], temperature [4], growth rate [5,6], stress [7,8], growth phase [9], and nutritional state [10]. Such mutation rate plasticity inspires the idea of "anti-evolution drugs," able to slow the evolution of antimicrobial resistance during the treatment of an infection [2,11–13]. Even small reductions in the mutation rate (2- to 5-fold) can have dramatic effects on the capacity of bacterial populations to adapt to antibiotic treatment, particularly when evolution is limited by mutation supply, as is the case for small pathogen populations [2].

**Data Availability Statement:** All relevant data are within the paper and its Supporting Information files.

**Funding:** RG is supported by Biotechnology and Biological Sciences Research Council (BBSRC) DTP BB/T008725/1 https://www.ukri.org/councils/bbsrc/. RK is supported by the UK Research and Innovation (UKRI) Future Leaders Fellowship MR/T021225/1 https://www.ukri.org/. PP is supported by Biotechnology and Biological Sciences Research Council (BBSRC) research grant BB/R007691/1 https://www.ukri.org/councils/bbsrc/. All sponsors played no role in study design, data collection and analysis, decision to publish, or preparation of the manuscript.

**Competing interests:** The authors have declared that no competing interests exist.

**Abbreviations:** BPS, base pair substitution; DAMP, density-associated mutation rate plasticity; DO, dissolved oxygen; DTPA, diethylenetetraaminepentaacetic acid; ODE, ordinary differential equation; PCR, polymerase chain reaction; ROS, reactive oxygen species.

Microbial mutation rates have an inverse association with population density across all domains of life; we have previously shown that 93% of otherwise unexplained variation in published mutation rate estimates is explained by the final population density [3]. This density-associated mutation rate plasticity (DAMP) is a distinct phenotype from stress-induced mutagenesis, which acts via independent genetic mechanisms [14]. Population density alters not only the rate but also the spectrum of mutations, with significantly higher rates of AT>GC transitions seen in low-density populations [15]. Density effects are likely relevant to natural populations given that population sizes and densities vary greatly, for example, *Escherichia coli* populations in host faeces can range in density by 5 orders of magnitude [16], and infections can be established by populations as small as $6 \times 10^3$ cells [17]. We therefore aim to mechanistically describe the widespread phenotype of DAMP.

In order to test potential mechanisms generating DAMP, we developed and systematically assessed a computational model connecting metabolism and mutagenesis in a growing *E. coli* population. This model generates the hypothesis that the key determinants of DAMP are the production and degradation rates of reactive oxygen species (ROS). Though molecular oxygen is relatively stable, it can be reduced to superoxide ($^\bullet O_2^-$), hydrogen peroxide ($H_2O_2$), and hydroxyl radicals ($HO^\bullet$). These "reactive oxygen species" are strong oxidants able to damage multiple biological molecules including nucleotides and DNA [18]. We tested the role of ROS in controlling DAMP by estimating mutation rate plasticity under different conditions of environmental oxygen and with genetic manipulations known to alter ROS dynamics. We find that the reduction in mutation rate at increased population density results from the population's increased ability to degrade $H_2O_2$, resulting in reduced ROS-associated mutagenesis. We show that this density effect is also experienced by cells deficient in $H_2O_2$ degradation when cocultured with wild-type cells able to detoxify the environment. Cross-protection from ROS has been previously demonstrated (e.g., [19]); however, the relevance of this cross-protection to mutation rates and in the absence of added $H_2O_2$ is novel. Mutation rates therefore depend not only on the genotype of the individual but also on the community's capacity to degrade $H_2O_2$.

## Results

### Initial computational model of nucleotide metabolism in a growing microbial population fails to reproduce mutation rate plasticity

To generate hypotheses for the mechanisms of DAMP, we constructed a system of ordinary differential equations (ODEs) that recapitulates the dynamics of metabolism, growth, and mutagenesis in a 1-ml batch culture of *E. coli* (Fig 1). The enzyme MutT, responsible for degrading mutagenic oxidised deoxy GTP [20], is essential in DAMP [3]; the ODE model is therefore focussed on guanine bases. In the model external glucose (***eGlc***) is taken up by a small initial *E. coli* population (***wtCell***). Internal glucose (***iGlc***) is then metabolised to produce ***dGTP***, ***ROS*** [21–24] and, largely, "other" molecules ("Sink" in Fig 1). ***dGTP*** is then either integrated into a newly synthesised DNA molecule (***DNA***) or it reacts with ***ROS*** to produce 8-oxo-2′-deoxyguanosine triphosphate (***odGTP***). In this model, non-oxidised ***dGTP*** always pairs correctly with cytosine, producing non-mutant DNA (***DNA***). In a second round of DNA replication, the guanine base is now on the template strand, cytosine is correctly inserted opposite producing new chromosomes (***wtCell***). ***odGTP***, if it is not dephosphorylated by MutT into *dGMP* (Sink), can either pair correctly with cytosine (becoming ***DNA***) or mis-pair with adenine (becoming ***mDNA***). When ***odGTP*** is inserted opposite adenine into DNA (***mDNA***), it may be repaired by the MutS or MutY proteins, converting the ***mDNA*** back to ***DNA***. As with DNA, mDNA undergoes a second round of DNA replication to be fixed in the genome as

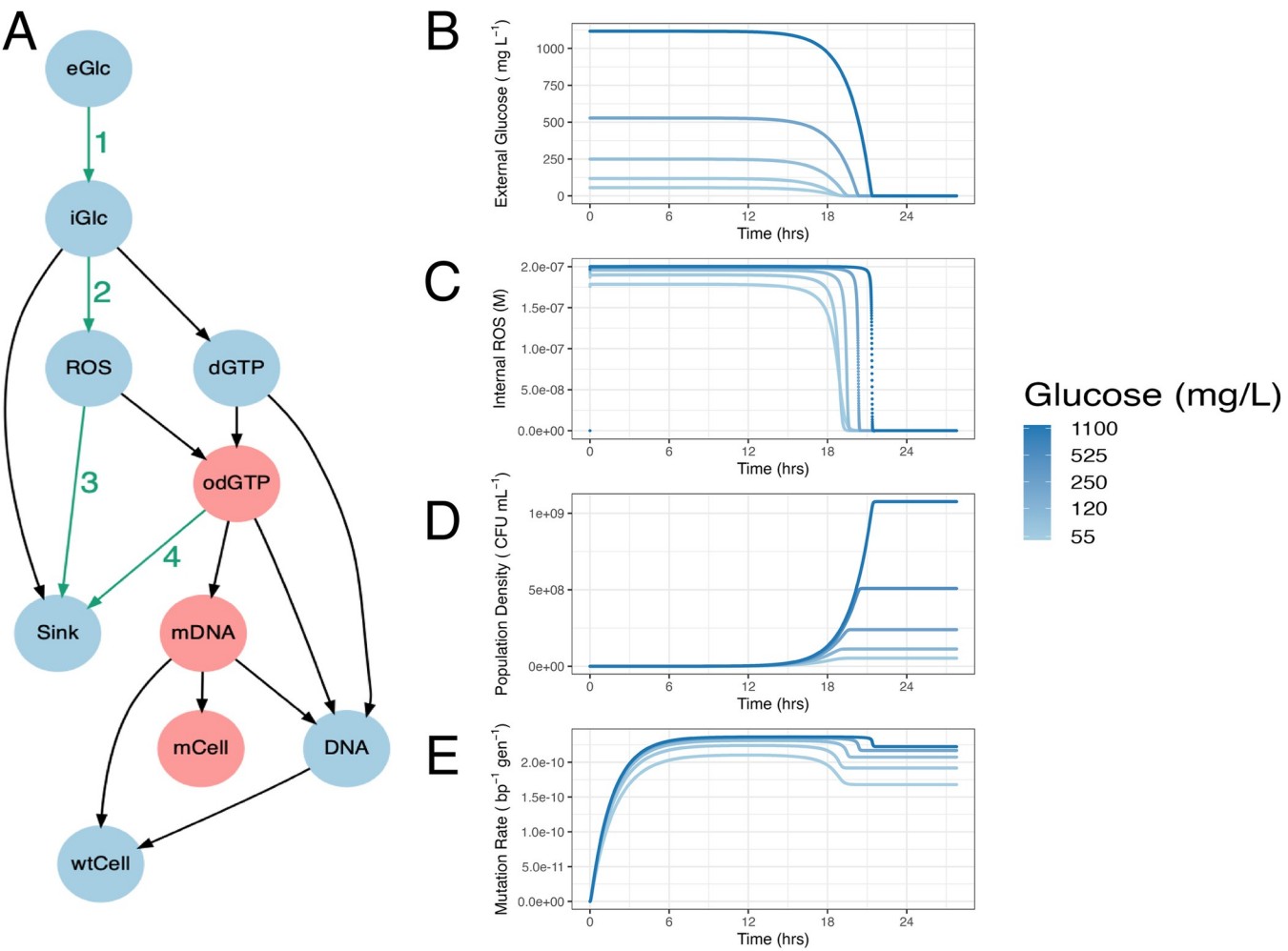

**Fig 1. Dynamical computational model of growth, metabolism, and mutagenesis in *E. coli*.** (A) Model structure connecting variables. Red variables indicate the pathway to mutagenesis; green numbered arrows indicate pathways targeted by model variants. This structure was represented in ODEs (Eqs 1–10, Methods), parameterised from the literature (Table 2), and simulated from appropriate starting values (Table 1) to give output shown in B–E. (B) Kinetics of eGlc, (C) molar concentration of ROS in the cytoplasm, (D) population density calculated as the number of genomes in the 1 ml culture (wtCell/GCperGen), and (E) mutation rate calculated as the ratio of mutated to total base pairs therefore representing the chance of a single base-pair mutating in a single division (generation); this is a cumulative measure of mutation rate in the sense that it considers all the mutations that have accumulated up to the given time, making it directly comparable to experimental measures of mutation rate. Panels B–E are plotted for 5 initial glucose concentrations (range 55 – 1,100 mg L$^{-1}$ as shown in legend), initial glucose concentration indicated by line colour. Raw data for panels A–E can be found in S1 Data. eGlc, external glucose concentration; ODE, ordinary differential equation; ROS reactive oxygen species.

mCell. Because these mutant base-pairs do not replicate, ***mCell*** measures the number of mutational events, referred to as "m" in mutation rate estimation [25]. The key output of interest is the mutation rate, which is defined as the number of mutant base pairs (***mCell***) divided by the total number of base pairs (***wtCell*** + ***mCell***). The model comprises 10 ODEs, one for each substance variable in Fig 1 (excluding "Sink"), plus ***cytVol***, the total population cytoplasmic volume within which all the reactions occur (Table 1, Eqs 1–10, Methods). These equations require 14 parameters (some of them composite, Table 2); the structure and parameter values are largely taken from the existing literature (for details, see Methods). Un-measurable parameters (notably the rate of ***dGTP*** oxidation to ***odGTP*** by ***ROS***, "O2") were set to give the observed mutation rate ($2 \times 10^{-10}$ mutations per base pair per generation, [26]) at a final population density of $3 \times 10^{8}$ CFU ml$^{-1}$, typical of 250 mg L$^{-1}$ glucose in minimal media. As with

**Table 1. Definitions and starting values for the 10 variables in ODE model A (Fig 1A).** For variables measured as a concentration, the volume within which this is calculated is given in the "region" column. wtCell and cytVol starting values equate to 2,175 cells (assuming 2357528 GC bp in the *E. coli* genome (strain MG1655, EBI Accession U00096.3)) and cell volume of $1.03 \times 10^{-12}$ ml [83]).

| Variable | Definition | Starting value | Units | Region |
|---|---|---|---|---|
| eGlc | External glucose | $3.1 \times 10^{-4}$ to $6.2 \times 10^{-3}$ (log spaced) | M | 1 ml culture |
| iGlc | Internal glucose | 0 | M | Cytoplasm |
| dGTP | Deoxyguanosine triphosphate | 0 | M | Cytoplasm |
| DNA | Guanine nucleotides in the newly synthesised strand | 0 | M | Cytoplasm |
| wtCell | Guanine nucleotides in the template strand | $8.5 \times 10^{-12}$ | M | 1 ml culture |
| ROS | Reactive oxygen species ($H_2O_2$) | 0 | M | Cytoplasm |
| odGTP | 8-oxo-2′-deoxyguanosine triphosphate | 0 | M | Cytoplasm |
| mDNA | odGTP nucleotides in the newly synthesised DNA strand | 0 | M | Cytoplasm |
| mCell | odGTP nucleotides in the template DNA strand | 0 | M | 1 ml culture |
| cytVol | Total cytoplasmic volume | $2.25 \times 10^{-9}$ | ml | NA |

ODE, ordinary differential equation; ROS reactive oxygen species.

most experiments demonstrating DAMP [3,27], final population density is controlled by varying initial external glucose. We initiated approximately 28 h simulations of 1 ml cultures with 2,175 cells (a small number, typical of fluctuation assays estimating mutation rate, S1 Fig), no internal metabolites and external glucose concentrations relevant to wet-lab experiments—across a log scale from 55 to 1,100 mg L$^{-1}$ (Table 1). The dynamics of external glucose, ROS, population size, and mutation rate for these simulations are shown in Fig 1B–1E, dynamics of all variables are included as Fig A2 in S1 Appendix.

This initial model (Fig 1, referred to as model A) creates an approximately linear log-log slope of 0.09 ± 0.016 (95% CI, Regression 1 (SI)) between final population and mutation rate (red line, Fig 2A, Regression 1 (SI)). We can compare the slope directly to in culture estimates of mutation rates in *E. coli*, which show strong DAMP, with a slope of −0.83 ± 0.13 (95% CI, grey dots and dashed line, Fig 2A, Regression 2 (SI)). Model A is therefore not describing the processes causing DAMP—the structure and/or the parameters used are either incomplete or fail to replicate biology for some other reason. To test whether inappropriate parameter values could be responsible for the lack of DAMP in model A, we simulated 50,000 parameter sets simultaneously varying all parameters randomly across 10% to 1,000% of their original value. These results were filtered as described in Methods and are plotted in Fig 2B (far left). This global sensitivity analysis showed the mutation rate plasticity, i.e., slope of model A to be very robust, with an interquartile range of 0.02 to 0.13 as shown by error bars in Fig 2B. All tested parameter sets gave a log-log linear slope of > - 0.06, suggesting that DAMP requires processes not represented in this initial model.

## ROS production and degradation are central to density-associated mutation rate (DAMP) plasticity in silico

While there was only limited variation in the relationship between mutation rate and population density, defining the slope of DAMP, in model A (Fig 2B), we can ask which model parameters are associated either with this slope variation or with variation in mutation rate itself across the set of models with all parameters simultaneously perturbed in the global sensitivity analysis (S2 Fig). The affinity of importers for glucose (Ks, part of reaction 1 in Fig 1A) had by far the closest association with the DAMP slope (Spearman's Rho $_{(DF = 3583)}$ = 0.91, $P < 2.2 \times 10^{-16}$, S2 Fig), whereas a group of parameters, including parameters controlling the

**Table 2. Parameter values and descriptions for all parameters used in model A.**

| Parameter | Value | Units | Description | Source |
|---|---|---|---|---|
| U1 | 2.66E-01 | $M^{-1}s^{-1}$ | Maximum uptake rate (Vmax) of eGlc | Fitted from a known value of Ks [92] and data on *E. coli* growth dynamics [93] (S13 Fig). |
| M1 | 2.69E-04 | $s^{-1}$ | Rate of dGTP synthesis from iGlc | This value was fitted to published data on *E. coli* growth dynamics [93] (S13 Fig). |
| Ks | 3.97E-05 | M | Michaelis Menten constant $K_s$: Concentration of glucose at which glucose uptake rate of 1/2 Vmax is achieved | $K_s$ measured as 7.16 µg/ml by [92]. |
| I1 | 6.90E-03 | $s^{-1}$ | Rate of dGTP incorporation into DNA opposite C | Fitted to give known cytoplasmic concentration dGTP in exponential growth phase *E. coli* of 92 µm in 0.4% glucose [94]. |
| D1 | 6.90E-03 | $s^{-1}$ | Rate of C pairing opposite incorporated G | Given the same value as I1 as the synthesis of new DNA (**DNA**) and new genomes (**wtCell**) should generally proceed at equal rates (this is violated during rapid exponential growth [95] but not included in this model). |
| O2 | 12.0 | $M^{-1}s^{-1}$ | Rate of dGTP oxidation to odGTP by ROS | Selected to give a mutation rate of $1.94 \times 10^{-10}$ base pair substitutions per nucleotide in 0.2% glucose minimal media [26]. |
| I2 | 2.53E-04 | $s^{-1}$ | Rate of odGTP incorporation into DNA opposite adenine | The relative efficiency of odGTP binding to A (I2) compared to G binding to C (I1) is $7.7 \times 10^{-8}$: $2.1 \times 10^{-6}$ (20), therefore, I2 = I1 * $(7.7 \times 10^{-8}/2.1 \times 10^{-6})$. |
| D2 | 2.00E-04 | $s^{-1}$ | Rate of C pairing opposite incorporated oG causing an AT>CG mutation | Rate of C pairing opposite an odGTP (D2) relative to CG (I1 and D1) is $6 \times 10^{-8}$: $2.1 \times 10^{-6}$ (20). |
| C1 | 2.8 | $s^{-1}$ | Rate of odGTP hydrolysis to odGMP by nudix hydrolase enzyme MutT (NudA) | Value taken from Kcat of MutT measured in vitro [96]. |
| C2 | 3.50E-04 | $s^{-1}$ | Rate of removal of adenine base incorporated opposite 8-oxodG in the genome by enzyme MutY | Value taken from Kcat of MutY measured in vitro as 0.021 min$^{-1}$ = $3.5 \times 10^{-4}$ sec$^{-1}$ ([97]). |
| R1 | 2.00E-04 | $s^{-1}$ | Rate of oG insertion into DNA opposite cytosine | Relative incorporation efficiency opposite C of odGTP:dGTP (R1) is 0.029 (20), therefore, R1 = I1 * 0.029. |
| S | 2.58E-02 | $s^{-1}$ | Rate of removal of adenine base incorporated opposite 8-oxodG in the genome by enzyme MutS | Fitted to known rate of mutation in *mutS* knockout of 40× wild type [98]. |
| r | 17.3 | $s^{-1}$ | Rate of ROS production from iGlc relative to dGTP production rate | Selected to give a known $H_2O_2$ production rate of 14 µm/second in 0.2% glucose minimal media normalised to cell volume [32]. |
| O3 | 5.60E+01 | $s^{-1}$ | Rate of ROS degradation through reactions other than dGTP oxidation. Primarily AhpCF/KatEG enzyme activity. | Fitted to give a standing ROS concentration of $1.9 \times 10^{-7}$ M, midpoint of known $1.3 \times 10^{-7}$ - $2.5 \times 10^{-7}$ M in LB [23]. |
| R2 | 2.53E-04 | $s^{-1}$ | Rate of adenine pairing opposite incorporated oG | The relative efficiency of odGTP binding to A (R2) compared to G binding to C (I1) is $7.7 \times 10^{-8}$: $2.1 \times 10^{-6}$ (20), therefore, R2 = I1 * $(7.7 \times 10^{-8}/2.1 \times 10^{-6})$. |
| Met1 | 1,545 | $s^{-1}$ | Stoichiometry of glucose conversion to dGTP for genome building (i.e., how many molecules of glucose are needed to produce 1 molecule of dGTP) | Fitted to published data from [3] of cell density as a product of glucose concentration (S14 Fig). |
| CellVol | 1.03E-12 | ml | Volume of one *E. coli* cell in minimal media growing in exponential phase | Mean value of 4 estimates of cell volume in exponential phase cells grown in minimal M9 media [83]. |
| molML | 6.02E+20 | molecules | Number of molecules per mL in a 1 M solution | One thousandth of Avogadro's constant ($N_A$). |
| GCperGen | 2357528 | GC basepairs | Number of GC basepairs per genome | *E. coli* Strain MG1655, EBI Accession U00096.3 |

eGlc, external glucose concentration; ROS reactive oxygen species.

rates of both **ROS** production (r, reaction 2 in Fig 1A) and **ROS** degradation (parameters O2 and O3, corresponding to reaction 3 in Fig 1A) had the closest association with the mutation rate (Spearman's Rho $_{(DF = 3583)}$ = 0.22, 0.21, and −0.21, respectively, all $P < 2.2 \times 10^{-16}$, S2 Fig). The parameter representing MutT activity (parameter C1, reaction 4, Fig 1A), found to be relevant in previous work on DAMP [3], was also in this group of parameters controlling mutation rate and so was also considered as candidate processes for further exploration. We hypothesise that the additional processes required to reproduce DAMP as observed in the lab

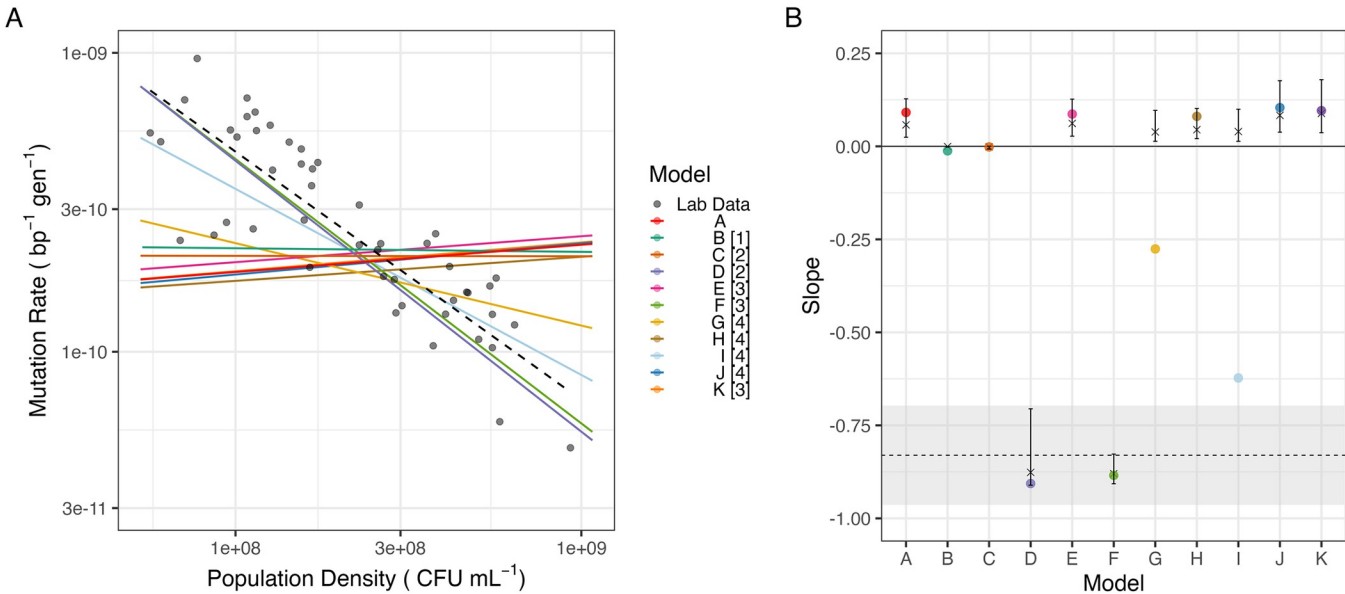

**Fig 2. Mutation rates in model variants.** (A) Solid coloured lines show fitted log-log linear relationship between final population density and mutation rate for models A to K (Regression 1 (SI)); numbers 1–4 in legend indicate the pathway targeted from Fig 1A. Black points and dashed line show lab data for *E. coli* wild-type BW25113 in glucose minimal media with a log-log linear regression fitted (Regression 2 (SI)). (B) Global sensitivity analysis; coloured points show slopes from baseline parameters (as in Fig 2A), and crosses and error bars show median and interquartile range of slope across $5 \times 10^5$ randomly perturbed parameter sets, models are coloured as in Fig 2A. Dashed line shows slope of lab data in Fig 2A (Regression 2), and grey area shows 95% CI on this slope. Raw data for panel A can be found in S2 Data, raw data for panel B (prior to filtering) can be found in S3 Data, and summary statistics as plotted can be found in S4 Data. eGlc, external glucose concentration; ODE, ordinary differential equation.

are associated with these reactions (numbered 1 to 4 in Fig 1A). We systematically tested each of these processes using structural variants to the model, explicitly modifying density dependence in biologically plausible ways. We thus use these models as a method of hypothesis generation, to determine which mechanisms may plausibly cause DAMP, with a view to testing these candidate mechanisms in the lab.

The slight increase in mutation rates seen as density increases in model A (a reversal of the negative association seen in the DAMP phenotype, therefore, referred to as "reverse DAMP") is the result of increased external glucose leading to increased internal glucose concentrations (S3 Fig). Since **ROS** production in model A is linearly related to internal glucose (Eq 7), this higher internal glucose results in higher mutation rates. It is therefore plausible that if glucose importer proteins are more expressed under low external glucose conditions, increasing the rate of reaction 1 (Fig 1A) at low glucose concentrations may increase mutation rates at low density. Introducing this model variant (model B, using Eq 1$_B$) does indeed remove model A's positive association between mutation rate and density but does not give the negative association observed in culture (model B slope = −0.01 ± 0.016 (95% CI), Fig 2, Regression 1 (SI)).

In model A, **ROS** are produced only by cellular metabolism (at a rate linearly related, with slope '$r \times M1$', to glucose metabolism); however, lab media also accumulates significant concentrations of $H_2O_2$ through photochemistry [28]. This is represented in model C by replacing reaction 2 (Fig 1A) with a constant **ROS** concentration in the system (using Eq 8$_C$ rather than Eq 8 and Eq 3$_C$ rather than Eq 3) and in model D by a constant rate of **ROS** production (using Eqs 7$_{DA}$ and 7$_{DB}$ rather than Eq 7). Both models abolish model A's positive slope. However, while model C removes DAMP (slope = −0.001 ± 0.03 (95% CI), Regression 1 (SI)), model D introduces a strong negative slope similar to the laboratory data (slope = −0.91 ± 0.016 (95%

CI), Regression 1 (SI)). As model D introduces ROS to the environment (representing the photochemical production of $H_2O_2$ in lab media [28]), we model both internal (Eq $7_{DA}$) and external ROS (Eq $7_{DB}$) with passive diffusion across the membrane as in [29]. Interestingly, the DAMP produced by this model is dependent on membrane permeability with decreased permeability reducing the slope of DAMP (Fig A5 in S1 Appendix).

Decreasing mutation rates at higher population densities could also be the result of changes in cellular ROS degradation rates (reaction 3). We therefore created models where degradation is determined by the internal glucose concentration (model E) and by the population density (model F), replacing Eq 7 with Eqs $7_E$ and $7_F$, respectively. Of these 2, the first had very little effect (model E, slope = 0.09 ± 0.016 (95% CI), Regression 1 (SI)), whereas the second had a large effect, giving a strong slope similar to in culture measurements (model F, slope = −0.89 ± 0.016 (95% CI), Regression 1 (SI)).

Given that previous work has shown the action of MutT in degrading ROS-damaged dGTP (*odGTP*, Fig 1A) to be essential to DAMP [3], we explored models in which the rate of *odGTP* degradation by MutT (reaction 4, Fig 1A) is determined by the internal glucose (model G, using Eq $8_G$ rather than Eq 8), *odGTP* (model H, using Eq $8_H$ rather than Eq 8) or *ROS* concentration (model I, using Eq $8_I$ rather than Eq 8). None of these models consistently resulted in DAMP (Fig 2B): making MutT activity dependent on *odGTP* had very little effect at all (model H, slope = 0.08 ± 0.03 (95% CI), Regression 1 (SI)), whereas making MutT activity directly responsive to internal glucose or ROS concentration did reproduce some degree of DAMP slope (models G and I slopes −0.28 ± 0.016 and −0.62 ± 0.016, respectively (95% CI), Regression 1 (SI)). However, the DAMP slopes of models G and I are highly parameter dependent with the majority of parameter combinations in the global sensitivity analysis giving very little slope at all (Fig 2B).

Finally, we replaced model A's mass action dynamics with saturating Michaelis Menten kinetics for MutT activity (reaction 4, model J using Eq $7_J$ rather than Eq 7 [20]) and enzymatic degradation of $H_2O_2$ (reaction 3, model K using Eq $7_{KA}$ and Eq $7_{KB}$ rather than Eq 7, [29]). Neither of these modifications greatly affected the mutation rate response of the model to population density (slope = 0.10 ± 0.016 and 0.096 ± 0.016, respectively, (95% CI), Regression 1 (SI), Fig 2). Thus, across 11 biologically plausible model structures chosen as those most likely to affect mutation rate plasticity, only 2, D and F, affecting reactions 2 and 3, respectively, in specific ways, produced DAMP comparable to that observed in the laboratory (Fig 2A) and robust to parameter variations (Fig 2B). Further details of the behaviour of all ODE model variants are included in S1 Appendix.

We can use these model findings for hypothesis generation: Model A (without DAMP) only describes ROS production from metabolism, whereas Model D (with DAMP) modifies the initial model to have a constant rate of ROS generation, independent of the cell density. Model D is consistent with ROS production in the system being dominated by environmental sources at a constant rate. If DAMP is a result of such environmental ROS production, we would expect this phenotype to be absent under anaerobic conditions where external $H_2O_2$ production is negligible [28].

Model F, which gains DAMP relative to model A, describes an increased rate of ROS detoxification dependent on the population density. This reflects a system in which ROS detoxification is primarily occurring within cells. Here, ROS diffusion into cells from the environment is significant and therefore the environment is more efficiently detoxified by larger populations. If DAMP is a result of an increased environmental detoxification capacity in dense populations in this way, we expect strains deficient in ROS degradation not to show DAMP. We would further expect dense populations to show greater removal of environmental ROS than low-density populations.

We therefore go on to test these predictions in culture using fluctuation assays to estimate the mutation rate in batch cultures of *E. coli*.

## Environmental oxygen is necessary for DAMP in culture

To test the hypothesis (from model D) that DAMP is dependent on external oxygen, we estimated mutation rates of *E. coli* under anaerobiosis across a range of nutrient-determined final population densities, analysing the results using a linear mixed effects model (Regression 4 (SI)). We find that anaerobic growth results in a loss of the negative relationship between density and mutation rate, indeed mutation rates significantly increased with density (slope = 0.65 ± 0.41 (95% CI), Fig 3B, statistical tests in S1 Table, Regression 4 (SI)). We further test this relationship using a second wild-type strain (*E. coli* MG1655). Again, we see a loss of DAMP under anaerobiosis (slope = 0.14 ± 0.7 (95% CI), cf. slope = −0.43 ± 0.25 (95%CI), anaerobic and aerobic, respectively, Regression 4 (SI), S4 Fig). This supports the hypothesis arising from model D that when external ROS production is substantial (model D/aerobiosis), mutation rates fall with increasing final population size, while when external ROS production is not included (model A/anaerobiosis) mutation rates remain similar or increase slightly with higher cell densities.

As well as losing DAMP, cultures grown under anaerobic conditions show significantly reduced mutation rates (LR = 15.83, $P = 1 \times 10^{-4}$, Regression 4, S1 Table). This raises the question of whether high-density cultures display reduced mutation rates because of increased

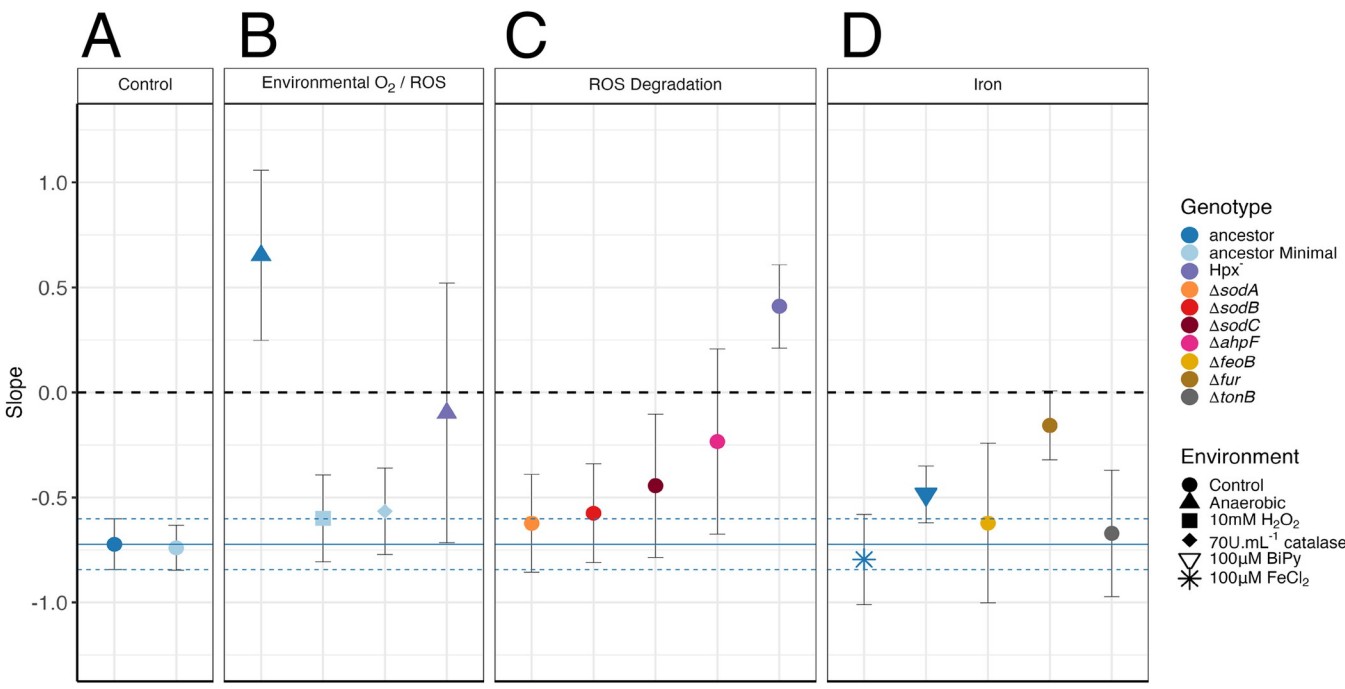

**Fig 3. Mutation rate responses to population density in culture under environmental and genetic manipulations.** Points show the slope of a log-log relationship between final population size and mutation rate (raw data shown in S5 Fig, Regression 4 (SI)), error bars show 95% CI on slope. Treatments shown are BW25113 ancestor (1,122 parallel cultures (pc) across 70 fluctuation assays (fa)); ancestor minimal media (974 pc, 61 fa); Δ*ahpF* (266 pc, 17 fa); Hpx⁻ (546 pc, 35 fa); ancestor anaerobic (168 pc, 11 fa); ancestor 10mM $H_2O_2$ (243 pc, 16 fa); ancestor 70U ml⁻¹ catalase (231 pc, 15 fa); Hpx⁻ anaerobic (105 pc, 7 fa); ancestor + chelator 2,2,Bipyridyl 100 μm (382 pc, 24 fa); ancestor + $FeCl_2$ 100 μm (210 pc, 13 fa); Δ*feoB* (210 pc, 13 fa); Δ*fur* (504 pc, 31 fa); Δ*tonB* (113 pc, 7 fa); Δ*sodA* (134 pc, 9 fa); Δ*sodB* (151 pc, 10 fa): Δ*sodC* (150 pc, 10 fa). Dashed line shows a slope of 0 (no DAMP); solid blue line shows the slope of BW25113 ancestor in rich media with dashed blue lines showing 95% CI on this estimate (Regression 4 (SI)). All experiments were conducted in dilute LB media unless stated "Minimal," in which case glucose minimal media was used. Raw data can be found in S5 data and summary statistics as plotted are in S1 Table. DAMP, density-associated mutation rate plasticity.

oxygen consumption compared with low-density cultures, resulting in a more anaerobic environment [30,31]. To address the hypothesis that the level of anaerobiosis depends on cell density, we measured dissolved oxygen concentrations in high- and low-density populations over the course of the growth cycle (S6 Fig). Although oxygen concentrations vary significantly between time points ($F = 22.5$, $df = 17$, $P = 3.7 \times 10^{-6}$), with an expected decrease during exponential growth, there is no difference in oxygenation between low- and high-density cultures in our system ($F = 1.2$, $df = 17$, $P = 0.29$). Furthermore, even where others have found a relationship between population density and oxygenation [31], this is highly nonlinear. Therefore, if DAMP were driven by oxygen availability we would expect differing DAMP slopes in subsamples of the data at high and low density. We find DAMP in wild-type BW25113 below and above a density of $1 \times 10^8$ CFU mL$^{-1}$ to show indistinguishable slopes ($-0.89 \pm 0.72$ ($N = 11$) and $-0.70 \pm 0.82$ ($N = 59$) ($\pm$SD), respectively; comparison t = $-0.80$, DF = 467, $P = 0.42$). Thus, while environmental oxygen is necessary for DAMP, its differential utilisation by high- and low-density cultures is unlikely to be its cause.

## Endogenous ROS degradation is necessary for DAMP in culture

The second ODE model able to reproduce DAMP (model F) introduces increased rates of ROS degradation with increasing population density. If DAMP is the result of active cellular ROS degradation, we would expect strains deficient in this trait to lack DAMP. The 2 alkyl hydroperoxide reductase subunits AhpC and AhpF are together responsible for the majority of $H_2O_2$ scavenging in aerobically growing *E. coli* [32]. The remaining $H_2O_2$ is degraded by the catalase enzymes HPI (*katG*) and HPII (*katE*) [33]. The role of catalases in $H_2O_2$ scavenging is much more significant at high $H_2O_2$ concentrations due to the higher Michaelis constants of these enzymes, whereas AhpCF is saturated at approximately 20 μm [33]. We therefore estimated DAMP in a version of the *E. coli* MG1655 strain lacking *ahpC*, *ahpF*, *katG*, and *katE* (Hpx$^-$, [21]). This quadruple deletion results in a complete loss of DAMP with no significant change in the mutation rate across densities (slope = $0.41 \pm 0.2$ (95% CI), Fig 3C, Regression 4 (SI)). Enzymatic degradation of $H_2O_2$ is thus essential to the DAMP phenotype, consistent with model F. Deleting only *ahpF* gives an intermediate DAMP phenotype (slope = $-0.23 \pm 0.44$ (95% CI), Fig 3C, Regression 4 (SI)) with significantly weaker DAMP than the wild-type (LR = 4.8, $P = 0.028$, Regression 4 (SI)), but still retaining stronger DAMP than Hpx$^-$ (LR = 7.4, $P = 0.0064$, Regression 4 (SI)), indicating that DAMP requires both catalase and alkyl-hydroperoxide reductase activity. In contrast, individual knockouts affecting superoxide rather than $H_2O_2$ (the superoxide dismutase genes *sodA*, *sodB*, and *sodC*, slope = $-0.62 \pm 0.23$, $-0.58 \pm 0.24$, and $-0.44 \pm 0.34$, respectively (95% CI), Fig 3C, Regression 4 (SI)), or adding environmental $H_2O_2$ or catalase (slope = $-0.6 \pm 0.2$, $-0.57 \pm 0.2$, respectively (95% CI), Fig 3B, Regression 4 (SI)) do not significantly disrupt the wild-type negative relationship between population density and mutation rate (S1 Table). It seems likely that the addition of extracellular catalase does not impact DAMP because in low $H_2O_2$ concentrations, such as those in lab media, it is alkyl hydroperoxide reductase (AhpCF) which plays a larger role than catalase in $H_2O_2$ degradation [33].

If the DAMP reproduced by model F is biologically realistic in this way, it requires that high-density populations, exhibiting reduced mutation rates, show greater efficiency at removing $H_2O_2$ from their environment than low-density populations. We measured external $H_2O_2$ in cultures after 24 h of growth in rich or minimal media and found high-density populations to achieve significantly lower $H_2O_2$ concentrations ($F_{28} = 24.3$, $P = 3.3 \times 10^{-5}$, Regression 7B (SI), S7A Fig); there was no significant effect of rich versus minimal media ($F_{26} = 0.77$, $P = 0.39$, Regression 7A (SI)). The log-log relationship between $H_2O_2$ and population density

($-0.33 \pm 0.1$, 95% CI, Regression 7B) is of a similar magnitude to the log-log relationship between mutation rate and population density ($-0.43 \pm 0.25$ and $-0.58 \pm 0.24$ in rich and minimal media respectively, 95% CI, Regression 4). The reverse pattern is seen in sterile media where increasing nutrient provision leads to increased $H_2O_2$ concentration ($F_{46} = 9.8$, $P = 3 \times 10^{-3}$, Regression 6 (SI), S7B Fig). This supports the hypothesis that, as embodied in model F, high-density populations detoxify external $H_2O_2$ better than low-density populations. Testing the expression dynamics of $H_2O_2$ degrading enzymes at high and low population density could clarify whether this greater $H_2O_2$ degradation capacity is purely the result of an increased population size (as in model D) or also reflects changed expression of enzymes such as AhpCF (as in model F).

## Cellular iron regulation is required for DAMP

Our model-guided hypothesis testing has shown that DAMP requires $H_2O_2$. Our models involve the direct effect of ROS on DNA; however, it is the reaction of free Fe(II) with $H_2O_2$ to produce mutagenic OH· radicals, Fenton chemistry, which is a major source of oxidative stress in *E. coli* [34,35]. These radicals are far more reactive and damaging to DNA than $H_2O_2$ itself, making iron critical to determining the amount of damage $H_2O_2$ causes [36]. If DAMP's dependence on $H_2O_2$, is the result of variable oxidative damage to DNA and nucleotides, we would expect this mutation rate plasticity to be perturbed by changes in cellular iron homeostasis. We first tested this using environmental manipulations of iron. However, the provision of $FeCl_2$ or starving cells of iron with a chelator (2,2-bipyridyl) has little effect on DAMP (Fig 3D and S1 Table). Nonetheless, we find that a deletant of *fur*, the master regulator of intracellular iron, results in an almost constant mutation rate across cell densities, with a significant reduction in DAMP compared to the BW25113 wild-type (slope = $-0.16 \pm 0.16$ (95% CI); wild-type slope comparison: $LR = 29.8$, $P = 4.9 \times 10^{-8}$, Regression 4 (SI), S1 Table). Although Fur is a regulator of many genes including ROS detoxification genes *katE/G* and *sodB/C*, it is Fur's central role as a negative regulator of multiple iron importers [37], which causes $\Delta fur$ strains to have an elevated internal redox-active iron pool [38] leading to increased oxidative stress and DNA damage ([39,40], S5 Fig). Knockouts of the iron importer genes *feoB* and *tonB*, which, if anything, reduce intracellular iron [41,42], do not lead to any change in mutation rate plasticity (Fig 3D and S1 Table), likely because regulators such as Fur may maintain iron homeostasis in the absence of these individual importers.

The critical contribution of iron to $H_2O_2$ stress is further demonstrated through whole genome sequencing of the Hpx⁻ LC106 strain used here. We find a 190-bp loss-of-function mutation in the iron importer *fecD* (all mutations listed in S2 Table). This may have allowed this Hpx⁻ strain to escape the positive feedback cycle that Hpx⁻ cells experience, in which higher $H_2O_2$ concentrations prevent Fur from effectively limiting iron uptake. More intracellular free iron then further exacerbates the damage done by the excess $H_2O_2$ [43,44]. It is likely that this loss-of-function mutation is an adaptation, during laboratory culture, to the loss of Fur functionality caused by the oxidation of intracellular iron. This raises the question of whether the lack of DAMP in Hpx⁻ arises not from the *katE/G ahpCF* knockout, but from this secondary mutation. To address this, we show that the independently derived Hpx⁻ strain BE007, without mutations in the *fecD* gene (S2 Table), also lacks DAMP ($X^2 = 0.808$, $df = 1$, $P = 0.369$) (S8 Fig).

## Wild-type cells restore DAMP in cells deficient in peroxide degradation

We have identified DAMP as requiring environmental $H_2O_2$ and endogenous $H_2O_2$ degradation, so that, with wild-type iron regulation, increased detoxification of environmental $H_2O_2$

leads to lower mutation rates at higher final cell densities. This understanding leads us to predict that the presence of wild-type cells should restore DAMP in the peroxidase and catalase-deficient Hpx⁻ strain. It has previously been shown that a wild-type population can provide protection against environmental $H_2O_2$ to cocultured Hpx⁻ cells, or similarly $H_2O_2$ sensitive $\Delta oxyR$ cells, through decreasing the peroxide concentration of the external environment [32,45]. To better distinguish Hpx⁻ and wild-type strains in a coculture, 2 nalidixic acid (Nal)-resistant strains of Hpx⁻ were independently created with the resistance conferred by point mutations in *gyrA* (D87G and D87Y). Coculturing these Hpx⁻$_{nalR}$ strains with wild-type BW25113 cells, the loss of DAMP via the Hpx⁻ mutation (Fig 3C) is phenotypically complemented by the wild-type cells. That is, Hpx⁻$_{nalR}$ mutation rate is significantly decreased in coculture with increasing population density either of the Hpx⁻ strain (Figs 4, S9, slope = $-0.93 \pm 0.5$ (95% CI), $X^2_{(DF=1)}$ = 11.7, $P = 6.4 \times 10^{-4}$, Regression 4 (SI)), or total population density (S10 Fig, Slope = $-1.4 \pm 1.06$ (95% CI), $t_{29} = -3.79$, $P = 7 \times 10^{-4}$, Regression 8 (SI)). This supports the hypothesis that DAMP is the result of reduced environmental $H_2O_2$ concentrations achieved by the local wild-type population. These results are also replicated by introducing an Hpx⁻ population to the ODE model D, highlighting the ability of this simple model to explain the DAMP phenotype (see Methods and S11 Fig).

Such mutation rate estimates in coculture could potentially be confounded by differential survival of rifampicin-resistant (RifR) mutants of Hpx⁻$_{nalR}$ when plated in a monoculture or a coculture. In order to test for any differences in mutant survival, we conducted a "reconstruction test" (as in S13 Fig of [27]); plating a predetermined number of Hpx⁻$_{nalR\&rifR}$ cells with a population of rifampicin susceptible Hpx⁻ or wild-type cells on the selective rifampicin agar. No significant difference in plating efficiency was seen between plating with Hpx⁻ versus low, medium, or high density of wild-type cells (S12 Fig; $LR_{df=6}$ = 11.7, $P = 0.07$, Regression 9 (SI)). Some difference in plating efficiency between the 2 Hpx⁻$_{nalR\&rifR}$ strains was observed (S12 Fig), this is likely due to the pleiotropic effects of different RifR resistance mutations in the *rpoB* gene [46,47]. The rifampicin resistance mutations in these strains were identified with Sanger sequencing of the *rpoB* rifampicin-resistance determining region and are given in S12 Fig. The specific protein sequence changes caused by these mutations have different known pleiotropic effects, as characterised in [47].

Although the wild-type strain reintroduces DAMP in Hpx⁻$_{nalR}$ it also causes an increase in total Hpx⁻$_{nalR}$ mutation rates (S10 Fig and S1 Table). This is potentially the result of out-competition by the wild-type strain leading Hpx⁻$_{nalR}$ growth to stop earlier in the culture cycle where, consistent with previous fluctuation assay results ([48], Chapter 5.4.3), our modelling leads us to expect higher mutation rates (Fig 1E and Fig A3 in S1 Appendix).

## Discussion

Using ODE modelling (Figs 1 and 2) to guide experiments in *E. coli* cultures, we have been able to predict and demonstrate the mechanisms behind the widespread phenomenon of reduced mutation rates at high microbial population densities (DAMP [3]). Genetic and environmental manipulations show that DAMP results from the improved degradation of $H_2O_2$ as the population density is increased (Fig 3). The reintroduction of DAMP in catalase/peroxidase deficient cells by coculture with wild-type cells (Fig 4) demonstrates the importance not only of a microbe's own population density in determining the mutation rate but also the density and genotype of coexisting populations. Our results demonstrate that mutation rates can be context dependent, through the degradation capacity of a community for mutagens including, but perhaps not limited to, $H_2O_2$.

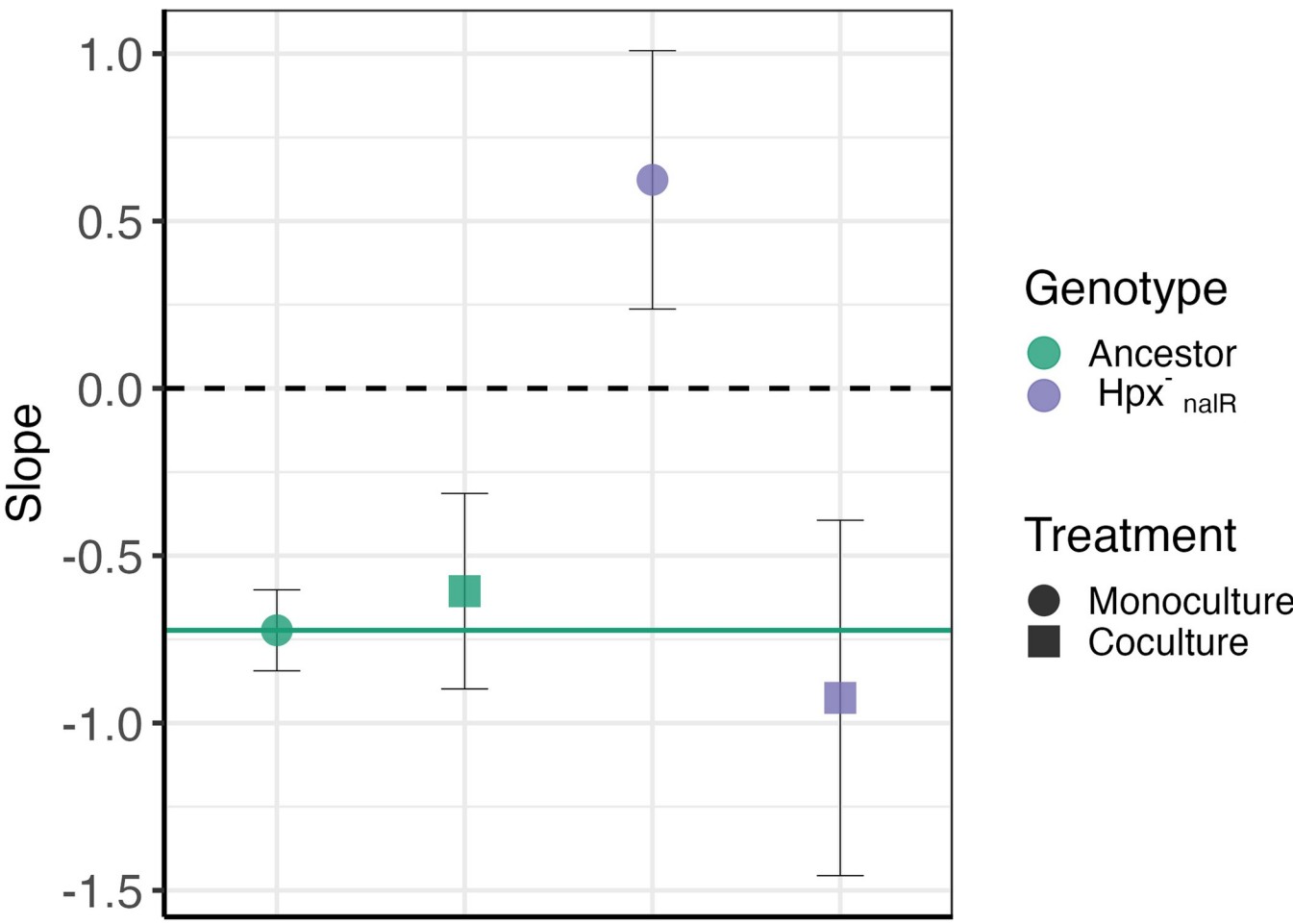

**Fig 4. Coculture with wild-type cells restores DAMP in cells deficient in peroxide degradation.** Points show log-log relationship between final population density of the focal strain and mutation rate fitted by Regression 4 (SI) (raw data shown in S9 Fig), error bars show 95% CI on slope. We found no significant differences between independent Hpx⁻_nalR strains; therefore, Hpx⁻ strains D87Y and D87G are combined in Hpx⁻_nalR. Treatments shown are: BW25113 ancestor (1,122 pc, 70 fa); BW25113 in coculture with Hpx⁻ (498 pc, 31 fa); Hpx⁻_nalR (388 pc, 24 fa); Hpx⁻_nalR in coculture with BW25113 (319 pc, 20 fa). Raw data can be found in S5 Data and summary statistics as plotted are in S1 Table. DAMP, density-associated mutation rate plasticity.

Increased population density provides protection against high levels of external $H_2O_2$ stress [19,45,49]. However, the concentrations of 100 μm to 1 mM applied in such studies is far beyond the range of known environmental concentrations, which is typically up to only 4 μm [50]. Here, we show that without any external input of $H_2O_2$ higher density populations detoxify environmental $H_2O_2$ more effectively over 24 h than low-density populations (S7 Fig). As well as improving survival under extreme $H_2O_2$ stress, previous work also finds mutation rates to decrease in cells protected by a higher density of neighbours able to detoxify the environment [45]. Here, we find that this mutation protection holds in the absence of external $H_2O_2$ application with the presence of higher density wild-type rescuers able to modify mutation rates in catalase/peroxidase-deficient cells (Fig 4). This interaction between 2 *E. coli* strains raises the question of whether similar interactions will be seen in mixed species communities such as human microbiomes where mutations can be critical for medically important traits, such as antimicrobial resistance [51,52].

This study, and previous work on DAMP [3,14,27], considers *E. coli* batch culture in which there is no renewal of media, meaning that peroxide detoxification is permanent. As media

inflow and outflow increase in a system, the ability of individual cells to detoxify ROS is decreased [53], it therefore seems possible that, increasing flow will be similar to transitioning from model D (fixed supply of environmental ROS, resulting in DAMP, Fig 2) to model C (fixed level of nucleotide oxidation, resulting in no DAMP, Fig 2). That would mean that the spatial structuring and resulting fluid dynamics of flow, which can be critical for bacterial competition and cross-protection [54,55], are also critical for mutation supply. Such factors vary greatly among natural environments, meaning that the effect of DAMP could be very different in low versus high through-flow environments (e.g., soil rather than water or lung rather than bladder). Tracing mutagenesis in single cells of spatially structured populations [56] has the potential to define the spatial scales and through flow conditions under which benefits from mutagen degradation are shared.

Our finding that oxygen is key to mutation rate plasticity is supported by mutation accumulation experiments showing that increased oxygen uptake is correlated with increased mutation rates [4]. However, existing literature is not agreed on this point—anaerobic fluctuation assay-like experiments report reduced mutation frequencies for resistance to multiple antibiotics [57], not unlike the reduced mutation rates we see in our anaerobic fluctuation assays (S5 Fig and S1 Table). Work assessing mutation rate by the accumulation of resistance mutants in chemostats also shows oxygen limitation to reduce mutation rates relative to carbon limitation [10]. In contrast, anaerobic mutation accumulation experiments instead report increased mutation rates [58,59]. This discrepancy is likely due to the change in mutational spectra caused by anaerobiosis: although overall mutation rates increase, base pair substitutions (BPSs) fall in frequency by 6.4 times [58] and it is such BPS which we modelled computationally and are often responsible for antibiotic resistance [60–62], particularly to rifampicin, the drug we used for our mutation rate estimates [61]. In line with our finding that iron and oxygen disruption are similarly able to abolish DAMP, iron and oxygen limitation produce similar mutational spectra [10]. The loss of DAMP in the $\Delta fur$ strain is perhaps due to higher intracellular iron levels producing a greater rate of $H_2O_2$ breakdown into DNA-damaging radicals before it can be detoxified, reflective of ODE model C in which a constant ROS burden is applied and no DAMP seen.

Our recent work suggests that AT>GC transitions are specifically elevated in frequency in low-density populations of E. coli [15]. Transitions are not generally regarded as a hallmark of ROS damage [63] and we do not explicitly include them in our modelling here. Nonetheless, the frequency of AT>GC transitions are the most elevated variant in naïve E. coli exposed to 1 mM $H_2O_2$ (as compared to cells primed by exposure to 0.1 mM $H_2O_2$) [64]. ROS stress imposed by either $H_2O_2$ treatment and $\Delta fur$ strains has been shown to elevate all BPS including density responsive AT>GC transitions. Future work, both with models and experiments is needed to clarify the relationship between ROS stress and all the elements of the mutational spectrum, to determine whether the collective $H_2O_2$ detoxification mechanism we have identified here is specific to the single mutational mechanism on which we built our modelling.

Mutation supply is a key evolutionary hurdle often limiting the adaptation of populations [1,65–67]. As mutation supply depends on population size, one might expect the supply of mutations, for instance to AMR, to be severely limited in small populations, such as the small number of cells forming an infectious propagule of E. coli [17]. Even when population size is sufficient to enable adaptation, mutation supply may have more subtle effects on the course of evolution, as demonstrated by the pervasive effects of mutational biases [68–70]. However, due to the action of DAMP in elevating mutation rates at low density, small populations can experience a very similar supply of mutations to large populations (as demonstrated in our data, S5 Fig). For E. coli at least, there is a limit to this effect as beyond intermediate densities (~$7 \times 10^8$

CFU $ml^{-1}$) the action of stress-induced mutagenesis causes mutation rates to rise, rather than fall, with increased density [14].

The collective protection from ROS we identify mirrors studies such as [71], demonstrating the importance of ROS control in microbial ecology. The dependence of DAMP on active cellular control of $H_2O_2$ concentrations, uncovered here, helps explain it is highly conserved nature. The evolution of cellular systems in an anaerobic world for approximately 1 billion years [72] means that all branches of life are similarly vulnerable to damage by ROS, leading to parallel effects of ROS damage across life [73], potentially including its population level control in DAMP. ROS defences such as those explored here, as well as excreted molecules including pyruvate [74–76], are therefore widespread in aerobic life. Although DAMP is highly conserved, it is notably not seen in *Pseudomonas aeruginosa* [3], despite this species being a close relative of *E. coli*. How DAMP is lost between such close evolutionary relatives remains an interesting open question. The current study suggests some speculative hypotheses: the formation of multicellular aggregates by *P. aeruginosa* [77] may make their experience of cell density more complex than the simple CFU per ml measure used here; the decreased permeability of *P. aeruginosa* [78] potentially decreases diffusion of $H_2O_2$ into the cytoplasm from external sources, which we predict to decrease DAMP (Fig A5 in S1 Appendix) and finally the greater ability of *P. aeruginosa* to detoxify environmental $H_2O_2$ [53], could mean that even low-density populations of *P. aeruginosa* can detoxify the environment as effectively as high-density populations, removing DAMP.

Population associations with mutation rate are widespread, including a significant negative relationship between the effective population size and mutation rates across vertebrates [79] as well as microbes [80]. Such patterns seem likely to be driven by the increased efficiency of natural selection against the deleterious effects of mutation in large populations (the drift barrier hypothesis, [81]), rather than any common mutagenic mechanism, as explored here, or any adaptive benefit. The broad reach of such non-adaptive explanations and the fact that the evolutionary effects of DAMP are yet to be explored means that any adaptive explanations should be approached with great caution. Nonetheless, in wild-type *E. coli*, the mutation rate has an inverse relationship not just with population density but also with absolute fitness, providing the greatest mutation supply to the most poorly performing populations [27]. Mutation supply also rises in the most nutrient-rich environments [14], potentially providing greater evolutionary potential where competition is most intense. Such plausible evolutionary benefits of DAMP could exist, even if the ultimate origins of its conserved mechanism lie not in selection for its indirect effects via mutation, but in the legacy, across domains of life, of the chemistry of the Great Oxidation Event [82].

## Materials and methods

### Ordinary differential equations and Model A

Models were created as coupled sets of ODEs for the change over time of relevant variables. These equations were largely parameterised from the literature, given appropriate starting values and simulated over time by integration with a solver, as described below.

All variables (Fig 1A and Table 1) are measured in molar concentration within the cytoplasm, aside from the volume of that cytoplasm (***cytVol***), measured in ml, and external glucose (***eGlc***) and number of growing cells (***wtCell*** and ***mCell***) which are measured as molar concentrations within the 1 ml batch culture. It is possible to convert between cytoplasmic and total metabolite concentrations through scaling by the cytoplasmic volume; this is calculated as the number of cells multiplied by a volume of $1.03 \times 10^{-9}$ µl per cell [83]. The reaction of ***dGTP*** with ***ROS*** creates oxidised dGTP (***odGTP***) which is then incorporated into DNA, creating

                                                              

AT > CG base pair substitution transversion mutations [26]. Mutations caused by **odGTP** may be avoided or repaired by the action of MutT, MutY, and MutS enzymes [84]. By dividing the number of mutant cells (**mCell**) by the total cell number (**mCell** + **wtCell**) at any point during the simulation, a mutation rate (bp$^{-1}$ generation$^{-1}$) across the simulation up to that point, can be calculated. Final population density is here manipulated by changing nutrient input, this also modifies a range of other timings and features of the culture cycle. We model growth with no death; though in practice death does occur in laboratory media, mortality rates are generally low [85]. Although DAMP is measured at approximately 28 h during the stationary phase where all metabolites are drained (Figs 1C, S3, and Fig A2 in S1 Appendix), we show that these slopes are also representative of DAMP during the exponential phase (16 h) (Fig A4 in S1 Appendix). This recapitulates in culture findings that DAMP slope is constant across growth phases in *E. coli* ([48], Chapter 5.4.3).

The uptake of glucose is described by saturating Michaelis Menten kinetics while the oxidation of **dGTP** is described as a bimolecular reaction dependent on the cytoplasmic concentrations of **dGTP** and **ROS**. All other steps are described by first-order mass action kinetics in which the rate equals the concentration of the reactant multiplied by a rate constant (Eqs 1–10). The model is parameterised from published enzymatic and culture data alongside our own wet lab data (Table 1).

R code to recreate all figures, models, and analysis relating to the ODE models is available as S1 Code. All models were simulated in R (V4.3.1) [86] using package deSolve (V1.36) [87]; logarithmic sequences were produced with emdbook (V1.3.13) [88]; data handling and plotting was done using the tidyverse (V2.0.0) [89] and magrittr (V2.0.3); and parallel computing was done using parallel (V4.3.0), doParallel (V1.0.17), and foreach (V1.5.2). Linear mixed models were fitted to lab data with nlme (V3.1-162) [90], and plots formatted and coloured using cowplot (V1.1.1), gridExtra (V2.3), ggeffects (V1.3.1) [91], and RColorBrewer (V1.1-3). Fig 1A was made using R package DiagrammeR (V1.0.10).

$$\frac{deGlc}{dt} = -U1\ wtCell\ \frac{eGlc}{eGlc + Ks} \tag{1}$$

$$\frac{diGlc}{dt} = \frac{U1\ wtCell\ \frac{eGlc}{eGlc + Ks}}{cytVol} - Met1\ M1\ iGlc \tag{2}$$

$$\frac{ddGTP}{dt} = M1\ iGlc - I1\ dGTP - dGTP \times ROS \times O2 \tag{3}$$

$$\frac{dDNA}{dt} = I1\ dGTP + C2\ mDNA + S\ mDNA + R1\ odGTP - D1\ DNA \tag{4}$$

$$\frac{dwtCell}{dt} = (D1\ DNA + R2\ mDNA) \times cytVol \tag{5}$$

$$\frac{dcytVol}{dt} = (D1\ DNA + R2\ mDNA) \times cytVol \times \frac{molML}{GCperGen}\ CellVol \tag{6}$$

$$\frac{dROS}{dt} = M1\ r\ iGlc - dGTP\ ROS\ O2 - O3\ ROS \tag{7}$$

$$\frac{dodGTP}{dt} = dGTP\ ROS\ O2 - C1\ odGTP - I2\ odGTP - R1\ odGTP \tag{8}$$

$$\frac{dmDNA}{dt} = I2\ odGTP - D2\ mDNA - C2\ mDNA - S\ mDNA - R2\ mDNA \tag{9}$$

$$\frac{dmCell}{dt} = D2\ mDNA\ cytVol \tag{10}$$

Equations 1–10: ODE equations for initial model (A).

## Model variants

### Model B—Glucose uptake increases at low eGlc

Original Michaelis Menten kinetics are removed as this reverses the intended effect.

$$\frac{deGlc}{dt} = -U1\ wtCell\ \frac{U2 - eGlc}{U2 - eGlc + K2} \tag{1B}$$

$$U2 = 6.7e - 3$$

$$K2 = 1.82e - 4$$

Both U2 and K2 are in Molar units.

$6.7 \times 10^{-3}$ is chosen as a value slightly higher than maximum eGlc so that the value of $\frac{U2-eGlc}{U2-eGlc+K2}$ can cover almost a full range of 0 to 1. This means that glucose uptake rate will increase from almost 0 to 100% of the measured uptake rate as the external glucose concentration falls. K2 is given as $1.82 \times 10^{-4}$ as this value produces the most negative DAMP slope achievable within the structure; values were tested from $1.82 \times 10^{-6}$ to $1.82 \times 10^{-2}$ (S1 Code).

### Model C—Constant dGTP oxidation, regardless of ROS concentration

By decoupling ROS concentration from dGTP oxidation there is no extra odGTP in cells grown to higher density, we expect this to prevent a positive DAMP slope. ROS no longer alters odGTP formation and therefore Eq 7 is removed from model C.

$$\frac{ddGTP}{dt} = M1\ iGlc - I1\ dGTP - dGTP \times ROSC \times O2 \tag{3c}$$

$$\frac{dodGTP}{dt} = dGTP\ ROSC\ O2 - C1\ odGTP - I2\ odGTP - R1\ odGTP \tag{8c}$$

$$ROSC = 1.8e - 7$$

ROSC in Molar units.

ROS is removed as a variable and replaced with a constant concentration of $1.8 \times 10^{-7}$, this is within the known internal ROS concentration of $1.3 \times 10^{-7}$ - $2.5 \times 10^{-7}$ M [23] and produces mutation rate of $1.93 \times 10^{-10}$ based on lab data and [26].

## Model D—Constant ROS production regardless of population density

By creating a situation in which ROS is produced in the media at a constant rate (e.g., [28]) and then diffuses into all present cells, higher density populations will expose each individual cell to less ROS. We expect this to create a negative DAMP slope as high-density populations will be able to maintain external, and therefore also internal, peroxide concentrations at lower levels.

$$\frac{dROS}{dt} = kdiff \ (ROS_{external} - ROS) - dGTP \ ROS \ O2 - O3 \ ROS \tag{$7_{DA}$}$$

$$\frac{dROS_{external}}{dt} = ROSC2 + \frac{cytVol}{1 - cytVol} kdiff \ (ROS - ROS_{external}) \tag{$7_{DB}$}$$

$$ROSC2 = 6e - 11$$

$$O2 = 40$$

ROSC2 in Molar units.

ROSC2 defines the number of millimoles of hydrogen peroxide produced in the media each second, this can then diffuse into the cells. The chosen value of $6 \times 10^{-11}$ creates an $H_2O_2$ production rate at 78% of that expected from [32] and an altering O2 to 40 restores the mutation rate to 95.5% of that expected from [26].

Permeability coefficient, diffusion coefficient, and cell surface area are taken from [29] to calculate the diffusion coefficient as follows:

$$kdiff = \frac{permeability \times surface \ area}{volume} = \frac{1.6 \times 10^{-3} cms^{-1} \times 1.41 \times 10^{-7} \ cm^2}{3.2 \times 10^{-12} \ cm^3} = 70$$

kdiff is in is in Sec$^{-1}$ units.

## Model E—ROS removal dependent on internal glucose

We expect greater rates of ROS removal to lead to lower rates of GTP oxidation, and therefore, lower mutation rates. If ROS is more able to be degraded when resources are abundant this may produce DAMP.

$$\frac{dROS}{dt} = M1 \ r \ iGlc - dGTP \ ROS \ O2 - O3 \ ROS \frac{iGlc}{iGlc + C3} \tag{$7_E$}$$

$$C3 = 1.5e - 4$$

C3 is in Molar units.

C3 is adjusted to produce known mutation rate of $1.98 \times 10^{-10}$ base pair substitutions per nucleotide in 0.2% glucose minimal media [26].

## Model F—ROS removal dependent on population density

We expect direct control of ROS degradation by population density to allow cells in higher density populations to avoid mutations more efficiently. The expression $\frac{MolML}{GCperGen} wtCell$ calculates the population density. The rate of AhpCF+KatEG catalysed ROS degradation is calculated in this model as this population density multiplied by numeric constant "C3a," replacing

"O3."

$$\frac{dROS}{dt} = M1 \; r \; iGlc - dGTP \; ROS \; O2 - ROS \frac{MolML}{GCperGen} wtCell \; C3a \tag{$7_F$}$$

$$C3a = 3.5e - 6$$

C3a is in Sec$^{-1}$ units.

C3a of $3.5 \times 10^{-6}$ is chosen to reproduce the mutation rate of $2.05 \times 10^{-10}$ base pair substitutions per nucleotide.

## Model G—MutT activity up-regulated by internal glucose

MutT activity is known to be essential to DAMP and so density-dependent MutT activity is a candidate DAMP mechanism. iGlc accumulates at higher levels in cells growing to high density, we expect high MutT activity in these cells to lead to a reduced mutation rate due to MutT cleaning of odGTP.

$$\frac{dodGTP}{dt} = dGTP \; ROS \; O2 - C1 \frac{odGTP}{1 - \frac{iGlc}{C3G}} - I2 \; odGTP \tag{$8_G$}$$

$$C3G = 2.6e - 3$$

$$O2 = 70$$

C3G is in Molar units.

$2.6 \times 10^{-3}$ is selected as a number slightly higher than the maximum iGlc achieved (approximately 0.0023), this prevents MutT activity levels from falling below 0. O2 is refitted to 70 to restore desired mutation rate.

## Model H—MutT activity up-regulated by odGTP

If MutT activity is actively up-regulated to degrade odGTP at a higher rate upon exposure to higher odGTP concentrations, then we expect cells grown in higher glucose, with higher internal metabolite concentrations, to have a greater ability to evade mutations caused by odGTP.

$$\frac{dodGTP}{dt} = dGTP \; ROS \; O2 - C1 \frac{odGTP}{1 - \frac{odGTP}{C3b}} - I2 \; odGTP \tag{$8_H$}$$

$$C3b = 8e - 10$$

C3b is in Molar units.

$8 \times 10^{-10}$ is selected as a number slightly higher than the maximum odGTP achieved, this prevents MutT activity levels from falling below 0.

## Model I—MutT activity up-regulated by ROS

Reasoning and value selection as in Models G/H.

$$\frac{dodGTP}{dt} = dGTP \; ROS \; O2 - C1 \frac{odGTP}{1 - \frac{ROS}{C3c}} - I2 \; odGTP \tag{$8_I$}$$

$$C3c = 1.98e - 7$$

$$O2 = 130$$

C3c is in Molar units.

## Model J—Michaelis Menten MutT kinetics

Michaelis Menten kinetics describe saturating, enzyme catalysed reactions. In this situation, reaction rates proceed slower at low substrate concentrations rising to an asymptote at maximum reaction rate. As with models G/H/I, we expect this to reduce mutation rates by increasing MutT activity in high-density populations with greater internal metabolite concentrations.

$$\frac{dodGTP}{dt} = dGTP\ ROS\ O2 - C1\ odGTP\frac{odGTP}{odGTP + Kt} - I2\ odGTP \tag{$8_J$}$$

$$Kt = 4.8e7$$

$$O2 = 6.36e - 4$$

Kt is in Molar units and is the Michaelis Menten Km value.
Kt value given by [20], O2 is then titrated to restore mutation rate as in [26].

## Model K—Separated activity of ahpCF and katEG genes + limited diffusion of ROS across the plasma membrane

$$\frac{dROS}{dt} = M1\ r\ iGlc - dGTP\ ROS\ O2 - \frac{kAhp\ ROS}{ROS + kmAhp} - \frac{kKat\ ROS}{ROS + kmKat}$$
$$- kdiff\ (ROS - ROS_{external}) \tag{$7_{KA}$}$$

$$\frac{dROS_{external}}{dt} = kdiff\,(ROS - ROS_{external})\left(\frac{cytVol}{1 - cytVol}\right) \tag{$7_{KB}$}$$

As in [29], the activity of alkylhyrdoperoxidase and catalase proteins are separated to allow for their specialisations to low and high $H_2O_2$ concentrations, respectively. Michaelis Menten and Vmax constants are as follows:

$$[Ahp]k_{cat}^{Ahp} = kAhp = 6.6e - 4$$

$$kmAhp = 1.2e - 6$$

$$[Kat]k_{cat}^{Kat} = kKat = 4.9e - 1$$

$$kmKat = 5.9e - 3$$

Km in Molar units and [Conc]$k_{cat}$ in M.sec$^{-1}$units.

All diffusion parameters shared with model D are defined in the same way. $H_2O_2$ production rate and standing concentration are restored to expected values by altering the value of r:

$$r = 175$$

## Model D Hpx⁻ coculture

To assess the expected DAMP of an Hpx⁻ population cocultured with wild-type *E. coli*, we add a second population to model D which differs from the wild type only in that the rate of ROS degradation by AhpCF/KatEG, "O3," is set to 0. In this model, the populations of Hpx⁻ and wt cells interact only through shared **eGlc** and a shared pool of **ROS_external**. The starting values of all variables are as in Table 1 and the starting population of both cell populations are set as in Table 1. As in model D, ROS originates from an exogenous source and can then diffuse into cells. When ROS diffuses into the Hpx⁻ cells it is not degraded, it is only when it diffuses into wt cells that ROS can be removed from the system. Eq 11 describes eGlc (external glucose), Eqs 12–20 describe wt cell metabolism, Eq 21 describes the external ROS pool, and Eqs 22–30 describe the Hpx cell metabolism. All equations are included below:

$$\frac{deGlc}{dt} = -U1\ wtCell\frac{eGlc}{eGlc + Ks} - U1\ wtCelLhpx\frac{eGlc}{eGlc + Ks} \tag{11}$$

$$\frac{diGlc}{dt} = \frac{U1\ wtCell\frac{eGlc}{eGlc+Ks}}{cytVol} - Met1\ M1\ iGlc \tag{12}$$

$$\frac{ddGTP}{dt} = M1\ iGlc - I1\ dGTP - dGTP\ ROS\ O2 \tag{13}$$

$$\frac{dDNA}{dt} = I1\ dGTP + C2\ mDNA + S\ mDNA + R1\ odGTP - D1\ DNA \tag{14}$$

$$\frac{dwtCell}{dt} = (D1\ DNA + R2\ mDNA) \times cytVol \tag{15}$$

$$\frac{dcytVol}{dt} = (D1\ DNA\ cytVol + R2\ mDNA) \times cytVol \times \frac{molML}{GCperGen}\ CellVol \tag{16}$$

$$\frac{dROS}{dt} = -dGTP\ ROS\ O2 - O3\ ROS - kdiff\left(ROS - ROS_{external}\right) \tag{17}$$

$$\frac{dodGTP}{dt} = dGTP\ ROS\ O2 - C1\ odGTP - I2\ odGTP - R1\ odGTP \tag{18}$$

$$\frac{dmDNA}{dt} = I2\ odGTP - D2\ mDNA - C2\ mDNA - S\ mDNA - R2\ mDNA \tag{19}$$

$$\frac{dmCell}{dt} = D2\ mDNA\ cytVol \tag{20}$$

$$\frac{dROS_{external}}{dt} = ROSC2 + \frac{cytVol}{1 - cytVol - cytVol_{hpx}} kdiff(ROS - ROS_{external})$$
$$+ \frac{cytVol\_hpx}{1 - cytVol - cytVol\_hpx} kdiff(ROS\_hpx - ROS_{external}) \tag{21}$$

$$\frac{diGlc\_hpx}{dt} = \frac{U1\ wtCell\_hpx \frac{eGlc}{eGlc+Ks}}{cytVol\_hpx} - Met1\ M1\ iGlc\_hpx \tag{22}$$

$$\frac{ddGTP\_hpx}{dt} = M1\ iGlc\_hpx - I1\ dGTP\_hpx - dGTP\_hpxROS\_hpx\ O2 \tag{23}$$

$$\frac{dDNA_{hpx}}{dt} = I1\ dGTP_{hpx} + C2\ mDNA_{hpx} + S\ mDNA_{hpx} + R1\ odGTP_{hpx} - D1\ DNA\_hpx \tag{24}$$

$$\frac{dwtCell\_hpx}{dt} = (D1\ DNA\_hpx + R2mDNA\_hpx) \times cytVol\_hpx \tag{25}$$

$$\frac{dcytVol\_hpx}{dt} = (D1\ DNA\_hpx + R2\ mDNA\_hpx) \times cytVol\_hpx \times \frac{molML}{GCperGen} CellVol \tag{26}$$

$$\frac{dROS\_hpx}{dt} = -dGTP\_hpx\ ROS\_hpx\ O2 - 0\ ROS - kdiff(ROS\_hpx - ROS_{external}) \tag{27}$$

$$\frac{dodGTP\_hpx}{dt} = dGTP\_hpxROS\_hpx\ O2 - C1\ odGTP\_hpx - I2\ odGTP\_hpx$$
$$- R1\ odGTP\_hpx \tag{28}$$

$$\frac{dmDNA\_hpx}{dt} = I2\ odGTP\_hpx - D2\ mDNA\_hpx - C2\ mDNA\_hpx - S\ mDNA\_hpx$$
$$- R2\ mDNA\_hpx \tag{29}$$

$$\frac{dmCell\_hpx}{dt} = D2\ mDNA\_hpx\ cytVol\_hpx \tag{30}$$

## Global sensitivity analysis

For each parameter within each model, 50,000 values between 10% and 1,000% of the baseline value (Table 2), spaced evenly along a log scale were tested. The set of values for each individual parameter were then independently shuffled so that no parameters were correlated with one another, allowing for substantial exploration of the available parameter space. Of these 50,000 parameter sets, some encountered fatal errors in the ODE solver and so did not produce a DAMP slope estimate, the number of parameter sets run without fatal error is shown in Table 3 as "complete." Results were filtered for the following criteria: **(1)** Stationary phase is reached in all glucose conditions (defined as an average increase of less than 1 cell per 10 s

**Table 3. Counts of completed and filtered simulations from 50,000 parameter sets produced for global sensitivity analysis for each model variant.** "Complete" column lists the number of these parameter sets that were able to be simulated without fatal error from the ODE solver. "Filter" columns list how many parameter sets remained after filtering as described and numbered above.

|   | Complete | Filter 1 | Filter 2 | Filter 3 | Filter 4 | Filter 5 |
|---|---|---|---|---|---|---|
| A | 49971 | 29080 | 17179 | 17179 | 12318 | 12261 |
| B | 33274 | 17402 | 9422 | 9422 | 6858 | 5444 |
| C | 49988 | 29202 | 16735 | 16735 | 13513 | 13496 |
| D | 49204 | 28612 | 16310 | 16310 | 9715 | 9682 |
| E | 49555 | 27890 | 16793 | 16791 | 12078 | 10267 |
| F | 49982 | 29107 | 16717 | 16717 | 10117 | 10115 |
| G | 23311 | 9416 | 4013 | 4013 | 2934 | 2730 |
| H | 49900 | 8489 | 3116 | 3116 | 2265 | 2210 |
| I | 22832 | 9787 | 4447 | 4447 | 3546 | 3334 |
| J | 49994 | 27533 | 16277 | 16277 | 12158 | 11741 |
| K | 45558 | 21487 | 12099 | 12099 | 7789 | 7365 |

ODE, ordinary differential equation.

time step across the last 1,000 time steps (2.7 h) of the simulation); **(2)** final population size $>1 \times 10^7$ and $<1 \times 10^{10}$ at every glucose condition; **(3)** final population size increases with each increase in glucose concentration; **(4)** mutation rate $>2 \times 10^{-12}$ and $<2 \times 10^{-8}$ at all glucose conditions; and **(5)** log-log relationship between mutation rate and final population size is substantially linear (defined by R-squared $>0.5$). After this filtering, the following number of parameter sets was retained for each model (Table 3).

## Strains used in this study

The parent of the Keio collection is *E. coli* strain BW25113 (F-, Δ(*araD*-araB)567, Δ*lacZ*4787 (:: *rrnB*-3), λ-, *rph*-1, Δ(*rhaD-rhaB*)568, *hsdR*514). *E. coli* Hpx⁻ LC106 mutant is Δ*ahpCF'* kan:: Δ*ahpF* Δ (*katG*17::Tn10)1 Δ (*katE*12::Tn10)1 [21]. *E. coli* Hpx⁻ strain BE007 is from Benjamin Ezraty as described in [99]. *E. coli* single-gene knockouts Δ*fur*, Δ*feoB*, Δ*tonB*, and Δ*ahpF* are sourced from the Keio collection [100]. *E. coli* K-12 strain MG1655 is from Karina B. Xavier. Nalidixic acid-resistant strains Hpx⁻ (*gyrA* D87Y) and Hpx⁻ (*gyrA* D87G) were isolated from independent fluctuation assays of the original Hpx⁻ strains on 30 mg L⁻¹ nalidixic acid selective plating.

Strains Δ*fur*, Δ*feoB*, Δ*tonB*, Δ*ahpF*, Hpx⁻, Hpx⁻ₙₐₗᵣ(D87Y), and Hpx⁻ₙₐₗᵣ(D87G) were sequenced to 30× depth by MicrobesNG to verify gene deletions. Lack of KatE activity in Hpx⁻ was verified by covering a colony on TA agar with 30% $H_2O_2$ with no bubbles of oxygen observed (as in [33]); the MG1655 wild-type was used as a positive control. Mutations were identified using breseq version 0.36.0 [101,102] with bowtie2 version 2.4.1 and R version 4.2.0 and are listed in S2 Table. For Hpx⁻ strains, the reference genome used was the *E. coli* K-12 MG1655 genome [[103], NCBI accession U00096.3]. For Keio knockout strains, the reference genome used was the *E. coli* K-12 BW25113 genome [[104], NCBI accession CP009273.1], with additional annotations for insertion (IS) element regions to improve the calling of mutations related to IS insertion (modified Genbank format file as file S1 in [1]).

## Media

We used Milli-Q water for all media, all chemicals are supplied by Sigma-Aldrich unless stated otherwise. LB medium contained: 10 g of NaCl (Thermo Fisher Scientific), 5 g of yeast extract

(Thermo Fisher Scientific) and 10 g of tryptone (Thermo Fisher Scientific) per litre. DM medium contained 0.5 g of $C_6H_5Na_3O_7 \cdot 2H_2O$, 1 g of $(NH_4)2SO_4$ (Thermo Fisher Scientific), 2 g of $H_2KO_4P$ and 7 g of $HK_2O_4P \cdot 3H_2O$ per litre; 100 mg $L^{-1}$ $MgSO_4 \cdot 7H_2O$ (406 μmol) and 4.4 μg $L^{-1}$ thiamine hydrochloride were added to DM after autoclaving. Selective tetrazolium arabinose agar (TA) medium contained 10 g of tryptone, 1 g of yeast extract, 3.75 g of NaCl, and 15 g bacto agar per litre; after autoclaving 3 g of arabinose and 0.05 g of 2,3,5-triphenyl-tetrazolium chloride were added per litre, this was supplemented with freshly prepared rifampicin (50 μg $ml^{-1}$) or nalidixic acid (30 μg $ml^{-1}$) dissolved in 1 ml of methanol or 1 M NaOH, respectively, when required. For all cell dilutions, sterile saline (8.5 g $L^{-1}$ NaCl) was used.

## Fluctuation assays

Fluctuation assays were conducted as described in [105]. Briefly, initial growth of glycerol stocks in LB was carried out for 4 h for all strains aside from Hpx⁻ which was grown for 7 h due to its reduced growth rate. A dilution factor of 1,000× was then used for transfer to overnight cultures. Overnight acclimatisation was carried out in DM supplemented with 3.5% LB or 250 mg $L^{-1}$ glucose with nutrient type matching that of the fluctuation assay. The density achieved in the assay was manipulated by growth in varying nutrient conditions, either 2% to 5% LB diluted in DM or 80 to 1,000 mg glucose $L^{-1}$. Manipulation of density via nutrient provision potentially confounds density with both nutrient environment and growth rate; the effects of these factors have been shown to be separable with a distinct effect of density ([106], Chapter 4). Selective plates were prepared 48 h before use and stored for 24 h at room temperature followed by 24 h at 4°C. All strains were plated on rifampicin selective media.

Anaerobic conditions were produced by incubating the 96 deep well plates in an airtight 2.6 L container with 1 Anaerogen 2.5 L sachet (Thermo Scientific). The Anaerogen sachet rapidly absorbs oxygen and releases $CO_2$ creating anaerobic conditions. Aerobic plates of matching design were grown in an identical container ventilated with 8 × 4 mm diameter holes without an Anaerogen sachet. In these plates, 2 to 4 wells in each 96-well plate contained DM supplemented with 2.5% LB, resazurin, and *E. coli* MG1655, leaving space for fluctuation assays of 15 to 16 parallel cultures. On removing the 96-well plates from incubation the resazurin absorbance at 670 nm was measured; this quantifies the change from pink resorufin (aerobic cell growth) to clear dihydro resorufin (anaerobic cell growth), thus providing an objective measure of anaerobiosis (S15 Fig).

During coculture fluctuation assays between BW25113 wild-type and Hpx⁻ both strains were grown up in LB for 4 to 7 h, then in 3.5% diluted LB with DM overnight and then diluted into cultures of approximately $1 \times 10^3$ CFU $ml^{-1}$ as above. Some combination of these 2 initial cultures was then mixed in each parallel culture ranging from an Hpx⁻:wild-type ratio of 1:1 to 124:1 (recorded in S1 Data file as "Mut_to_WT_ratio"). Plating of these cultures on TA or TA +Rif agar enabled the Ara+ (white) Hpx⁻ colonies and the Ara- (red) wind-type colonies to be distinguished. For assays using NalR Hpx⁻ strains, selective plating was done on TA+Rif+Nal plates and so only Hpx⁻ mutants and not wild-type mutants were counted, Nt was determined for both strains using plating on both TA+Nal and TA. Due to amino acid synthesis defects, Hpx⁻ cells cannot be cultured in glucose minimal media and so all cocultures were conducted in dilute LB media [34].

In order to test for any differences in survival of Hpx⁻ (NalR) grown in monoculture VS with differing densities of BW25113 wild-type cells, we conducted a reconstruction test (S12 Fig). A known quantity of Hpx⁻ (NalR + RifR) cells were plated with one of the following treatments: sterile DM, Hpx⁻ (5%LB overnight growth), wild-type (2.5%LB overnight growth),

wild-type (3.5%LB overnight growth), and wild-type (5% overnight growth). Raw data from the reconstruction test is available as S11_data.xlsx.

Though fluctuation assays allow for high-throughput and low-cost estimates of mutation rate, they classically come with some important assumptions to consider [25]. For example, the assumption that resistance markers will be selectively neutral is not reasonable in practice [47]. Fortunately, this can be accounted for with the estimation of fitness cost which can then be accounted for in the estimation of mutational events using R package flan (V0.9). We find estimations of DAMP with the co-estimation of the individual fitness cost in each assay or with the application of the median mutant:wild-type fitness ratio estimation (median = 0.59; fitness cost estimates shown in S16 Fig, regression 3) to have no effect on our conclusions. Specifically, in regression 4 (SI), re-running the model using either of these approaches to account for genotypes having different competitive fitness makes no difference to whether DAMP is inferred (i.e., categorising each treatment as DAMP, no DAMP, or reverse DAMP; S1 Table) for all treatments. In this study, we allow flan to co-estimate fitness along with mutational events ($m$) for each assay. Occasionally, this model failed to converge on estimates, in these cases average fitness effects were estimated from a model fitted to all successful estimates (Regression 3 (SI)) and then used to estimate $m$ from the data with this predetermined fitness effect of mutation. It is also possible to avoid issues of mutant fitness effects by using the $p_0$ method of estimation [25,107] in which parallel cultures are simply divided into those with or without any viable mutants. However, this method is more restrictive as only in assays in which parallel cultures both with and without growth have been observed can the method be applied; it is also subject to more error on estimates than maximum-likelihood methods [25,108]. Reanalysing our data with the $p_0$ method shows DAMP to exist in almost all of the same treatments as in the original analysis (S17 Fig), discounting any effect of mutant fitness costs on our conclusions. Another potentially unrealistic assumption of the fluctuation assay is that there will be no death; this too is possible to account for using the tools provided in R package flan. We find that introducing a death rate of 25%, beyond what would be expected under our conditions which lack added stressors [85], cause no consistent or substantial changes in DAMP slope (S18 Fig).

## Sanger sequencing of *rpoB* mutations

PCR amplification of the rifampicin-resistance determining region (bp 1328–2235) of *rpoB* in rifR mutants (S12 Fig) was carried out as described in [15]. Rifampicin-resistant mutants isolated from a fluctuation assay were grown in LB to exponential phase and stored in 18% glycerol at −80°C. Mutants were revived by streaking of 1 µl of this glycerol stock onto LB agar plates. DNA was diluted by touching a colony with a pipette tip, submerging this pipette tip in 25 µl nuclease-free water, and diluting 1 µl of that solution in a further 9 µl of nuclease-free water. Approximately 1 µl of diluted DNA was added to 24 µl of master mix containing 1.25 µl of both forward and reverse 100 µm primer stock, 5 µl buffer, 5 µl MgCl$_2$, 0.25 µl high-fidelity DNA polymerase, and 11.25 µl nuclease-free water.

Polymerase chain reactions (PCRs) were carried out to amplify the rifampicin-resistance determining region of the *rpoB* gene for Sanger sequencing, using forward primer 5′-ATGA-TATCGACCACCTCGG-3′ and reverse primer 3′-TTCACCCGGATACATCTCG-5′ [15]. PCR was run with the following protocol: (i) initial denaturation (98°C for 5 min); (ii) denaturation (98°C for 10 s); (iii) annealing (55°C for 30 s); (iv) extension (72°C for 1 min); (v) repeat steps 2 to 4 for 35 cycles; (vi) final extension (72°C for 5 min); and (vii) hold at 4°C. PCR product verification was performed by gel electrophoresis carried out on 1% agarose TAE gel with 0.1% SybrSafe stain. PCR products were submitted to Source BioScience for PCR

Product clean-up and sequencing (Source BioScience, Cheshire). Both reverse and forward primers were submitted for each sample as sequencing primers.

Using Unipro UGENE v49.0, downloaded from https://ugene.net/, sequences of the *rpoB* gene of the mutants were aligned to a reference *rpoB* genome nucleotide sequence, of the strain *E. coli* K-12 substr. MG1655, obtained from EcoCyc version 27.5 to identify point mutations. BLAST was used to identify amino acid changes created by the mutations [109].

### Hydrogen peroxide measurement

External hydrogen peroxide is measured using the Amplex UltraRed (AUR)/Peroxidase assay as described in [28]. All reagents were dissolved in 50 mM dibasic potassium phosphate. Diethylenetetraaminepentaacetic acid (DTPA) and AUR solutions were corrected to pH 7.8 with HCl or NaOH. Reactions containing 660 μl 1 mM DTPA, 80 μl filter sterilised sample solution, and 20 μl 0.25 mM AUR were mixed by vortexing before transferring 141 μl to 3 wells of a clear bottomed black 96-well plate. Fluorescence was measured at 580 nm excitation, 610 nm emission before and after the injection of 7.5 μl horseradish peroxidase (0.25 mg ml$^{-1}$) to each well, net fluorescence was calculated as initial fluorescence subtracted from final fluorescence. $H_2O_2$ concentration was estimated by calibration to standard solutions of 5 and 20 μm $H_2O_2$ (Regression 5 (SI)). Because of background levels of fluorescence, some predicted concentrations were negative, this was accounted for by taking the absolute value of the lowest prediction and adding this to all predictions. The range of $H_2O_2$ concentrations we observed is in good agreement with similar measurements in the literature (e.g., Fig 6B in [28]).

### Dissolved oxygen (DO) measurement

Dissolved oxygen was measured in 10-ml cultures grown in 50-ml falcon tubes; DAMP has been shown to be present in *E. coli* grown under these conditions (see *E. coli* cultures plated on nalidixic acid in the second figure of [3]). Tubes were either loosened ½ turn with the lid secured with a small piece of tape or screwed on fully though no difference was evident between these treatments. Each starting culture of 40 ml was split between 4 tubes of 10 ml sampled at 0, 5, 7, and 24 h (sampling is destructive as the head of the DO probe cannot be confirmed to be sterile). DO was measured by submerging the probe in the given sample for 45 s to allow the measurement to stabilise. Following DO measurement, each culture was appropriately diluted and plated to determine population density by CFU. Dissolved oxygen analyser DO9100 was purchased from BuyWeek.

### Statistical analysis

All statistical analysis was executed in R (V4.3.1) [86] using the nlme (V3.1-162) package for linear mixed effects modelling [90]. This enabled the inclusion within the same regression of experimental factors (fixed effects), blocking effects (random effects), and factors affecting variance (giving heteroscedasticity). R package car (V3.1.2) [110] was used to carry out Chi-squared tests comparing slope to a null-hypothesis of 1 (S1 Table). In all cases, log$_2$ mutation rates were used. Details of all regression models are given S1 supplementary statistics along with diagnostic plots and ANOVA tables for each model. The code and data to reproduce the main text figures are given in the accompanying R scripts S1 and S2 Code files, and supplementary data files S1–S14 data, respectively. Column header definitions for S1–S14 data are given in S3 Table. Standard deviation on estimates of *m* is calculated as in [108]. The same R packages were used for parallel computing, data handling, and plotting as for the ODE modelling, with the addition of plyr (V1.8.8), ggbeeswarm (V0.7.2), and gridExtra (V2.3).

## Supporting information

**S1 Fig. Distribution of initial population size across all fluctuation assays.** Mean = 3,537, median = 3,000. Low population size is desirable in order to maximise the number of generations considered and to reduce the chances of resistant mutants being present in the starting population ("jackpot cultures"). Raw data can be found in S5 Data.
(TIFF)

**S2 Fig. Global sensitivity analysis of Model A.** Left-hand side (A) shows the absolute rank correlation, as quantified by Spearman's Rank Correlation Coefficient, between each parameter and the slope of DAMP, parameters are ordered from least to most correlation from left to right. Right-hand side (B) shows the equivalent information for the correlation between parameter values and mutation rate (at 250 mg L$^{-1}$). Positive correlations are shown in green while negatively correlated parameters are shown in orange. Black borders show significant rank correlation ($P < 0.05$). Note the different y axis limits and x axis order on the left- VS right-hand side. Raw data can be found in S3 data, Spearman's rank correlation coefficient statistics, and associated $p$-values can be found in S6 Data.
(TIFF)

**S3 Fig. Dynamics of internal glucose over time in model A simulated at 5 log-spaced glucose concentrations from 55 to 1,100 mg L$^{-1}$.** Higher levels of initial external glucose provision (point colour) lead to higher levels of internal glucose (y-axis). Raw data can be found in S1 Data.
(TIFF)

**S4 Fig. Slope of log-log relationship between population density (CFU ml$^{-1}$) and mutational events (ml$^{-1}$) in wild-type strain MG1655 under aerobic and anaerobic conditions.** Pink circle = MG1655 rich media anaerobic (173 pc, 11 fa); green triangle = MG1655 minimal media aerobic (273 pc, 17 fa); pink triangle = MG1655 rich media aerobic (285 pc, 18 fa). Orange line and shaded area shows DAMP for BW25113 in rich media as in Fig 3 with 95% CI. Raw data can be found in S5 Data and summary statistics as plotted are in S1 Table.
(TIFF)

**S5 Fig. Individual assay data underlying Fig 3.** Final population density is plotted against mutational events per ml on a log-log scale. Dashed lines show the null expectation of a constant mutation rate (i.e., slope = 1), the y intercept for the dashed lines is arbitrary. Coloured lines are fitted slopes from mod3 (S1 Supplementary Statistics file), line gradients with 95% CI shown in Fig 3. Treatments shown are BW25113 ancestor (1122 parallel cultures (pc) across 70 fluctuation assays (fa)); ancestor minimal media (974 pc, 61 fa); ΔahpF (266 pc, 17 fa); Hpx$^-$ (546 pc, 35 fa); ancestor anaerobic (168 pc, 11 fa); ancestor 10 mM H$_2$O$_2$ (243 pc, 16 fa); ancestor 70U ml$^{-1}$ catalase (231 pc, 15 fa); Hpx$^-$ anaerobic (105 pc, 7 fa); ancestor + chelator 2,2, Bipyridyl 100 μm (382 pc, 24 fa); ancestor + FeCl$_2$ 100 μm (210 pc, 13 fa); ΔfeoB (210 pc, 13 fa); Δfur (504 pc, 31 fa); ΔtonB (113 pc, 7 fa). Raw data can be found in S5 Data and summary statistics as plotted are in S1 Table.
(TIFF)

**S6 Fig. Dissolved oxygen concentration over time.** (A) O$_2$ concentration is plotted as a function of time in wild-type BW25113 cultures grown in 2% LB (low density–blue points and lines) or 5% LB (high density–red points and lines); 50-ml tubes contain 10 ml of culture with lids attached by a small piece of tape and either screwed on tight (triangles) or loosened one half turn (circles). Solid lines connect mean values for half turn loosened samples and dashed lines connect points for tight samples. While there is consistent variation in oxygen

concentration over time, there is no consistent variation among nutrient treatments. (B) Density measured at each time point alongside dissolved oxygen measurement. Raw data can be found in S7 Data.
(TIFF)

**S7 Fig. Effects of population density and nutrient level on $H_2O_2$.** Left-hand side (A) shows the log-log relationship between population density and external $H_2O_2$ in cultures of MG1655 after 24 h of incubation. Rich media is 2/5% LB diluted in DM, minimal media is 80/1,000 mg $L^{-1}$ glucose in DM. Population density is estimated from the optical density (OD 1 = $2.37 \times 10^9$ cells $ml^{-1}$ calculated from OD measurements taken alongside fluctuation assays). Lines of best fit are from regression 7B (SI). Right-hand side (B) shows the $H_2O_2$ concentration after 24 h incubation in rich or minimal media; sterile or with wild-type MG1655, Regression 6 (SI); error bars show 95% CI. The interaction effect between nutrient level (low versus high) and presence of a culture (Sterile Media versus Wild-type), where external peroxide decreases with nutrients increases in the presence of a culture but increases without one, is significant ($F_{DF=46} = 9.8$, $P = 3 \times 10^{-3}$, Regression 6 (SI)). Raw data can be found in S8 Data.
(TIFF)

**S8 Fig. DAMP in a second independent Hpx- strain.** Left-hand plot shows raw data used to calculate DAMP slope in Hpx⁻ LC106 (546 pc, 35 fa), Hpx⁻ BE007 (149 pc, 10 fa), and their ancestor MG1655 (285 pc, 18 fa). Right hand shows DAMP slope as fitted by regression 4. DAMP slope does not significantly differ between the 2 Hpx⁻ strains (LR = 0.71, DF = 110, $P = 0.4$); however, DAMP slopes do differ between MG1655 vs. LC106 (LR = 28.3, DF = 110, $P < 0.0001$) and MG1655 vs. BE007 (LR = 5.2, DF = 110, $P = 0.02$). Raw data can be found in S5 Data and summary statistics as plotted are in S1 Table.
(TIFF)

**S9 Fig. Individual assay data underlying Fig 4.** Final population density of the focal strain is plotted against mutational events per ml on a log-log scale. Dashed lines show the null expectation of a constant mutation rate with a slope of 1. Ancestor coculture measurements are taken in coculture with Hpx⁻, Hpx⁻ D87Y, and D87G are cocultured with ancestor BW25113. Lines are fitted slopes shown in Fig 4. BW25113 ancestor (1,106 pc, 69 fa); BW25113 in coculture with Hpx⁻ (498 pc, 31 fa); Hpx⁻nalR (388 pc, 24 fa); Hpx⁻nalR in coculture with BW25113 (319 pc, 20 fa). Raw data can be found in S5 Data and summary statistics as plotted are in S1 Table.
(TIFF)

**S10 Fig. Relationship between total population density and mutation rate in Hpx- with cocultured wild-type BW25113.** (A) Final population density (focal + coculture strain where relevant) is plotted against mutation rate on a log-log scale. Hpx⁻nalR monoculture (388 pc, 24 fa); Hpx⁻nalR in coculture with BW25113 (319 pc, 20 fa). Lines are fitted slopes shown from Regression 8 (SI). (B) Slope and 95% CI on the lines shown in LHS graph. Horizontal orange line shows the slope of the BW25113 ancestor in rich media (Regression 4 (SI), Fig 3). In monoculture Hpx⁻ mutation rates increase with total population density while in coculture the wild type restores a negative association between density and mutation rates (DAMP). Raw data can be found in S5 Data.
(TIFF)

**S11 Fig. Predictions of ODE model D, with Hpx⁻ + wild-type coculture alongside lab measurements.** Estimates from lab data are shown in red as in Fig 4. Estimates from ODE modelling are shown in blue, and 95% CI are included for all points; however, ODE model CI are too narrow to be visible. Lab data summary statistics shown can be found in S1 Table and raw data

in S5 Data; ODE model summary statistics shown can be found in S9 Data, raw data from ODE models can be found in S10 Data.
(TIFF)

**S12 Fig. Reconstruction test showing the plating efficiency of rifampicin-resistant Hpx⁻ (GyrA D87Y/RpoB S531F) and Hpx⁻ (GyrA D87G/Q513P)** when combined and plated in 1.25 ml with: sterile DM media (DM), Hpx⁻ overnight culture (hpx 5% LB), BW25113 low-density overnight (BW2.5% LB), BW25113 mid-density overnight (BW 3.5% LB), and BW25113 high-density overnight (BW 5% LB). Plating efficiency is calculated as the number of colonies counted divided by the number of colonies counted on non-selective TA agar plates without any additional treatment. Raw data can be found in S11 Data.
(TIFF)

**S13 Fig. Fit of model variant A to published data.** Lines show results of ODE model A simulated as described in methods, circles show data from [93] used to fit parameters U1 and M1. Left-hand panel shows the molar concentration of external glucose over time and right-hand panel shows *E. coli* cells per ml over time. Raw data for fitting can be found in S12 Data, data from ODE output can be found in S1 Data.
(TIFF)

**S14 Fig. Fit of linear mixed effects model relating final population density to initial glucose concentration.** Used to fit parameter Met1 in ODE models. Black points show published lab data from [3] on population density and glucose provision in *E. coli* MG1655 used to fit this regression. Black line and shaded area show fitted relationship and 95% confidence interval, respectively, of a mixed effects model accounting for random effects of experimental block and plate. Red stars show output, in final population density, from initial ODE model A under differing initial glucose concentrations. Raw data for fitting can be found in S13 Data, data from ODE output can be found in S1 Data.
(TIFF)

**S15 Fig. Reduction of resorufin to dihydro resorufin by anaerobic respiration results in reductions in absorbance at 670 nm verifying the anaerobic conditions during anaerobic fluctuation assays.** Each of 5 blocks is shown as a separate facet; within each block 2 sets of paired fluctuation assays (A and B) were conducted in aerobic and anaerobic conditions, for each of these sets 2–4 measurements of resorufin/dihydro resorufin absorbance were taken after 24 h of growth. Raw data can be found in S5 Data.
(TIFF)

**S16 Fig. Fitness effects of resistance mutations where fitness is co-estimated with mutational events.** Boxplots shown for each treatment with colour representing genotype. Vertical lines inside boxes represent the median for that treatment, with the boxes depicting the interquartile range. The black vertical line at a fitness effect size of 1 represents neutral fitness effects. This data is used to fit regression 3. Raw data can be found in S14 data.
(TIFF)

**S17 Fig. DAMP is mostly seen in the same set of treatments using maximum likelihood or $p_0$ estimation methods.** Treatments in which 8 or more fluctuation assays can be analysed by the $p_0$ method are shown. Colour indicates treatment identity. The only treatment to change category is Hpx⁻ strain LC106 which moves from "no DAMP" with ML estimates to "reverse DAMP" with $p_0$ estimates; this does not refute our conclusions that Hpx⁻ strains display no negative association between mutation rate and population density. All Hpx⁻ points are strain

LC106 unless indicated as BE007. Raw data used can be found in S5 Data.
(TIFF)

**S18 Fig. An assumption of 25% death has little effect on the estimation of DAMP slope.**
Solid lines indicate a slope of 0 (no DAMP), dashed line shows identical slope values for both
estimates. All treatments remain in the same category (DAMP, no DAMP, or reverse DAMP).
Raw data used can be found in S5 Data.
(TIFF)

**S1 Table. Slope estimates with associated Chi-Squared tests from Regression 4 (SI).** Slope
indicates the log-log relationship between population density and mutational events per ml
minus 1 (1 is subtracted to make interpretation simpler as a constant mutation rate is now
defined by a slope of 0 rather than a slope of 1). slope_CI95 indicates that a 95% confidence
interval on the slope estimate will be slope ± slope_CI95. pValue is calculated from a Chi-
Squared test (DF = 1) comparing the original slope value to the Null Hypothesis that the slope
of the given treatment = 1 (slope = 1 when mutation rate is constant with respect to population
density); therefore, in treatments in which the slope significantly differs from 1, we have
observed density associated mutation rate plasticity. FA and PC list the number of fluctuation
assays and parallel cultures used in the analysis respectively. Plasticity shows if the treatment
has DAMP (a significant inverse relationship between population density and mutation rate),
reverse DAMP (a significant direct relationship between population density and mutation
rate), or is Constant (relationship between population density and mutation rate not signifi-
cantly different from the null expectation of a constant mutation rate).
(CSV)

**S2 Table. Mutations, missing coverage, and new junction evidence for key strains in this
study as predicted by variant calling with breseq (run in default consensus mode).**
(XLSX)

**S3 Table. Descriptions for columns in S1–S14 Data files.**
(XLSX)

**S1 Code. R code necessary to recreate ODE modelling (Figs 1, 2, S2, S3, S11, S13, and S14).**
(R)

**S2 Code. R code necessary to recreate lab work analysis (Figs 3, 4, S1, S4–S10, S12, and
S15–S18).**
(R)

**S1 Supplementary Statistics. Details of statistical models used in this study.**
(PDF)

**S1 Appendix. Further details of ordinary differential equation model dynamics.**
(DOCX)

**S1 Data. Dynamics of ODE model A over time under 5 initial external glucose conditions.**
(CSV)

**S2 Data. Mutation rate across 5 densities for ODE models A–K.**
(CSV)

**S3 Data. Raw output from sensitivity analysis on ODE models A–K.**
(7Z)

**S4 Data. Summary statistics from sensitivity analysis on ODE models A–K.**
(CSV)

**S5 Data. Raw data from fluctuation assays.**
(CSV)

**S6 Data. Spearman's rank correlation coefficient comparing the value of each parameter in ODE model A to both mutation rate and DAMP slope with associated *P*-values.**
(CSV)

**S7 Data. Data collected from dissolved oxygen measurements of BW25113 cultures in 2% VS 5% LB.**
(TXT)

**S8 Data. Data from amplex ultra-red peroxide assays.**
(CSV)

**S9 Data. Estimated slope and standard deviation for coculture version of ODE model D with both wt BW25113 and Hpx⁻ populations.**
(CSV)

**S10 Data. Raw data used to estimate coefficients in S9_data.csv; final concentration of mutant base pairs and wild-type base pairs after approximately 27 h when simulating a coculture version of ODE model D with both wt BW25113 and Hpx⁻ populations.**
(CSV)

**S11 Data. Data collected from reconstruction test of Hpx-nalR&rifR strains plates with Hpx⁻ VS BW25113 at varying densities.**
(XLSX)

**S12 Data. Data available from [93] used to fit parameter U1.**
(TXT)

**S13 Data. Data available from [3] used to fit parameter Met1.**
(CSV)

**S14 Data. Estimated fitness cost of rifampicin resistance mutation co-estimated with mutational events using R package "flan".**
(CSV)

## Acknowledgments

Thanks to Danna Gifford for many insightful discussions during this study and for the improved BW25113 reference genome. Thanks also to The University of Manchester Research IT for their assistance and the use of the Computational Shared Facility. Thanks to MicrobesNG (http://www.microbesng.com) for their genome sequencing services. Thanks to Simon C. Andrews for providing the Hpx⁻ strain LC106, to Benjamin Ezraty and Patrice L. Moreau for providing the Hpx⁻ strain BE007, to Karina B. Xavier for providing the MG1655 wild-type strain, and to James Imlay for helpful correspondence.

## Author Contributions

**Conceptualization:** Rowan Green, Andrew J. McBain, Pawel Paszek, Rok Krašovec, Christopher G. Knight.

**Formal analysis:** Rowan Green.

**Funding acquisition:** Andrew J. McBain, Pawel Paszek, Rok Krašovec, Christopher G. Knight.

**Investigation:** Rowan Green, Hejie Wang, Carol Botchey, Siu Nam Nancy Zhang, Charles Wadsworth, Francesca Tyrrell.

**Methodology:** James Letton, Rok Krašovec, Christopher G. Knight.

**Resources:** James Letton.

**Supervision:** Andrew J. McBain, Pawel Paszek, Rok Krašovec, Christopher G. Knight.

**Visualization:** Rowan Green.

**Writing – original draft:** Rowan Green.

**Writing – review & editing:** Hejie Wang, Carol Botchey, Siu Nam Nancy Zhang, Charles Wadsworth, James Letton, Andrew J. McBain, Pawel Paszek, Rok Krašovec, Christopher G. Knight.

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
