## [Editor Report · Decision Letter 0]

12 Oct 2023

Dear Dr. Green, 

Thank you for submitting your manuscript entitled "Working together to control mutation: how collective peroxide detoxification determines microbial mutation rate plasticity." for consideration as a Research Article by PLOS Biology.

Your manuscript has now been evaluated by the PLOS Biology editorial staff, as well as by an academic editor with relevant expertise, and I am writing to let you know that we would like to send your submission out for external peer review as an Update Article. Please select this option when re-submitting. 

Once your full submission is complete, your paper will undergo a series of checks in preparation for peer review. After your manuscript has passed the checks it will be sent out for review. To provide the metadata for your submission, please Login to Editorial Manager (https://www.editorialmanager.com/pbiology) within two working days, i.e. by Oct 14 2023 11:59PM.

Kind regards,

Paula

---

Senior Editor

PLOS Biology

---

## [Decision Letter · Decision Letter 1]

14 Nov 2023

Dear Dr. Green,

Thank you for your patience while your manuscript "Working together to control mutation: how collective peroxide detoxification determines microbial mutation rate plasticity." was peer-reviewed at PLOS Biology. It has now been evaluated by the PLOS Biology editors, an Academic Editor with relevant expertise, and by several independent reviewers. 

In light of the reviews, which you will find at the end of this email, we would like to invite you to revise the work to thoroughly address the reviewers' reports.

As you will see below, the reviewers find the work interesting, but they raise several important concerns, many highlighted by reviewer #1. Some concerns require either more experiments or substantial revision (including some of the seemingly minor points raised by reviewers #2 and #3). You can also see the comments from the Academic Editor at the end of the letter. Please address all the concerns. 

Given the extent of revision needed, we cannot make a decision about publication until we have seen the revised manuscript and your response to the reviewers' comments. Your revised manuscript is likely to be sent for further evaluation by all or a subset of the reviewers.

**IMPORTANT - SUBMITTING YOUR REVISION**

*Re-submission Checklist*

*Published Peer Review*

*PLOS Data Policy*

*Blot and Gel Data Policy*

Sincerely,

Paula

---

Senior Editor

PLOS Biology

REVIEWS:

Reviewer #1: Modelling evolution of biofilm and antibiotic resistance.

Reviewer #2: Bacterial DNA repair and mutagenesis.

Reviewer #3: Christopher Marx. Ecology and evolutionary microbiology.

Reviewer #1: Summary

This Update article extends previous work on density-associated mutation rate plasticity (DAMP) with a computational model to generate hypotheses for the underlying mechanisms. Models that can reproduce DAMP are then explored experimentally to determine the underlying mechanisms of the observation of lower mutation rate in denser populations. Given the fundamental role of mutation rate variation in biology and that DAMP has been observed in taxonomically diverse species this work would be of interest to a wide range of biologists. The authors main conclusion is that the collective peroxide detoxification ability of microbial populations determines mutation rate plasticity. This is based on four main experiments.

A. No DAMP is observed in populations grown under anaerobic conditions

B. No DAMP is observed in E. coli populations deficient in the degradation of hydrogen peroxide

C. No DAMP is observed when the transcriptional regulator Fur is deleted which is expected to lead to higher iron levels in the cell and more ROS damage

D. Reduction of mutation rates in denser populations is restored in peroxide degradation-deficient cells by the presence of wild-type cells in a mixed population 

This study is a substantial update to the previous work both in terms of the model and the experimental work. It is not trivial to measure the slopes of mutation rates and the authors have conducted a very large number of fluctuation tests to produce data of high quality. I agree that most of the experimental results are consistent with the authors' interpretation that DAMP is caused by collective peroxide detoxification. However, I do not think that this is the only reasonable interpretation of the experimental results. The authors could do more to discuss and possibly rule out alternative explanations and perform a few additional experiments that could support or argue against their hypothesis. I am overall very positive to this high-quality work, but I cannot currently see that the authors´ conclusions are fully supported by the data presented.

Major comments

1. Dissolved oxygen will decrease rapidly as a population grows beyond a certain threshold (about 10^8cells/ml) in a poorly aerated culture such as in deep well plates. I have seen large differences in mutation rates for oxo-G mediated mutations in regular 96-well plates compared to tubes where aeration is better. When oxygen levels are lower then endogenous ROS production will be lower, which could lead to the observed reduction in mutation rates at high cell densities. A simple way to test this would be to see if there is DAMP in well-aerated cultures as well, for example baffled e-flasks with small volume and fast shaking. Under such conditions there should be high dissolved oxygen until at least 4x10^8 cells/ml (https://doi.org/10.1186/1475-2859-5-8). This hypothesis would also be consistent with the main experimental results but would have a different interpretation of the collective detoxification compared to reduced endogenous production of ROS. It could also be formulated as an alternative model where ROS production is decreased when population density increases above a certain level due to lower dissolved oxygen. This model could be experimentally tested as it would predict higher and similar mutation rates at low/medium cell densities as long a dissolved oxygen is high and lower mutation rates due to drastically reduced ROS production at high cell densities.

2. Fig S4 shows that the slopes were measured at quite different cell densities for different strains/conditions, which is problematic when they are compared only in terms of slopes in the main figures. For example, the catalase and additional of H2O2 experiments are done with populations of high density and the Hpx- mutant has much lower cell density when calculating the slope, presumably because of a growth defect. Obviously, this would be problematic if ROS degradation is directly related to cell density but also if the dissolved oxygen level is important for mutation rates.

3. A related point is that cell density here is determined by nutrient amount. This would also change other experimental conditions, for example how long a population has been in stationary phase. Would DAMP also be observed if mutation rate was measured at different time points during growth in the same concentration of nutrient? If this has already been confirmed in a previous study, it would be good to mention as it would strengthen the hypothesis that is cell density and not another correlated factor that is the key determinant of DAMP.

4. I find it problematic that the Hpx- strain has secondary mutations including a fecD deletion that is likely to have a major impact on mutation rate results as it will probably have lower iron levels in the cell and a lower mutation rate with could contribute to loss of DAMP. I realize that it would be a substantial effort to construct a new Hpx- mutant and that it is possible that a new mutant would also have secondary mutations. However, confirming key results with an independently constructed Hpx- strain would substantially strengthen the article and perhaps it would be possible to obtain a previously constructed strain for example that used in https://doi.org/10.1007/s00253-021-11169-2 or the ΔahpCF katG mutant used in (https://doi.org/10.1046/j.1365-2958.2001.02303.x). It might also be possible to repair the fecD gene or complement it by addition of a copy elsewhere in the genome. The conclusions rely heavily on this one strain and fecD and the other mutations makes it uncertain if the phenotypic effects are only due to the deletions of ahpAC, katE and katG.

5. In the computational model ROS is the only source of mutations. In model D the model is changed to constant ROS production. I find it very problematic that this ROS production is then simply divided between the cytoplasms of all cells (line 593). This will obviously lead to extremely high levels when there are few cells and low levels when there are many cells which explain why there is DAMP, but it is not realistic that all ROS produced in the environment is suddenly allocated to the inside of the cells present in the population.

6. In Model F, the other model showing DAMP, ROS degradation is directly determined by population density and as ROS is the only source of mutation this will obviously lead to a higher mutation rate at lower densities. Thus, this result is self-evident and there is no attempt to test this model experimentally for example by measuring expression of ahpCF, katE or katG at different cell densities.

7. Line 297-300. Adding environmental H2O2 or catalase do not disrupt the negative relationship between mutation rate and population density. I think this result needs to be discussed more and explained how it is consistent with the authors' hypothesis of how DAMP works. To me the interpretation would be that environmental H2O2 is not important. 

Minor comments

8. In the Variable table (line 493) WtCell and mCell is listed but in later equations the variables are named differently as dGcell (line 527) and dmGcell (line 537).

9. Ref 15 (line 78) reports increased AT->GC transitions in low density populations. Yet ROS associated mutagenesis through 8-oxo-G is known to give rise mainly to transversions with an increase in A->C transversions in a mutT mutant and increased G->T transversions in mutM mutY mutants. Sanger sequencing of rpoB for a selection of rifR mutants could be done to confirm that ROS associated mutations are more common at lower cell densities, which is fundamental to all models proposed here.

10. In lines 325-326 fur is described as the master regulator of intracellular iron. It would be helpful for the reader to know that Fur not only regulates genes related to intracellular iron but that it is a major regulator of >100 genes, which also included gene with other roles (https://doi.org/10.1038/ncomms5910) including ROS defense genes like katE and katG (https://regulondb.ccg.unam.mx/regulon/RDBECOLITFC00093).

11. Fig S6 shows relatively small differences in hydrogen peroxide concentration between low and high nutrient levels. Would this be enough to explain the difference in mutation rate?

Reviewer #2: The manuscript by Green et al. examines the mechanisms underlying the phenomenon that cellular mutation rates decrease as the density of cells in culture increases (DAMP = density associated mutation rate plasticity). This effect has previously been discovered and described by the labs that performed this study. The present manuscript goes a step further in testing specifically the hypothesis that the differences in mutation rates can be explained by the capacity of a culture for scavenging mutagenic reactive oxygen species (ROS, specifically H2O2). The authors developed a phenomenological model of ROS-induced mutagenesis dependent on the concentration of glucose in the culture media. Here, glucose concentration affected the rate of H2O2 production and the population density. This alone was insufficient to explain the DAMP effect, so the authors systematically varied parameters and model structure to recapitulate the effect. According to the story in the manuscript, this led to the realisation that ROS scavenging capacity of the culture varies with density, which in turn affects mutation rates. This was then tested via a series of mutation rate measurements with E. coli strain deficient in H2O2 detoxification mechanisms and iron regulation. While H2O2 scavenging-deficient cells showed no DAMP effect, a co-culture with wild-type cells restored the effect, showing that it is indeed a collective phenomenon. Overall, the article is interesting and provides clear insights into a complex phenomenon. I suggest addressing the following minor points:

1) Abstract: "in vivo mutation rate estimation". The phrase "in vivo" could be misleading as it might suggest mutation rate estimation of bacteria within hosts or in the environment, as opposed to "in vitro" lab culture conditions.

2) Introduction: "We show that this density effect is also experienced by cells deficient in H2O2 degradation when cocultured with wild-type cells able to detoxify the environment." Authors should cite here the classic Ma & Eaton 1992 paper showing that cel

---

## [Decision Letter · Decision Letter 2]

29 May 2024

Dear Dr Green,

Thank you for your patience while we considered your revised manuscript "Working together to control mutation: how collective peroxide detoxification determines microbial mutation rate plasticity." for publication as a Update Article at PLOS Biology. This revised version of your manuscript has been evaluated by the PLOS Biology editors, the Academic Editor and the original reviewers.

Based on the reviews, we are likely to accept this manuscript for publication, provided you satisfactorily address the remaining points raised by the reviewers and the following data and other policy-related requests.

IMPORTANT - please attend to the following:

a) Please N-terminally truncate your Title to ""Collective peroxide detoxification determines microbial mutation rate plasticity in E. coli"

b) Please address the remaining concerns from reviewer #1.

c) Please address my Data Policy requests below; specifically, we need you to supply the numerical values underlying Figs 1BCDE, 2B, 3ABCD, 4, S1, S2, S3, S4, S5, S6AB, S7, S8, S9, S10, S11, S12, S13, S14, S15, S16, S17, S18, either as a supplementary data file or as a permanent DOI’d deposition. We note that although the data may be in SupData 1-7, some of the files seem to be missing (SupData 2-4), and the relationship to specific Figs should be clarified.

d) Please cite the location of the data clearly in all relevant main and supplementary Figure legends, e.g. “The data underlying this Figure can be found in S1 Data” or “The data underlying this Figure can be found in https://doi.org/10.5281/zenodo.XXXXX”

We expect to receive your revised manuscript within two weeks. 

*Published Peer Review History*

*Press*

Sincerely,

Roli Roberts

Roland Roberts, PhD

Senior Editor

rroberts@plos.org

PLOS Biology

DATA POLICY:

Regardless of the method selected, please ensure that you provide the individual numerical values that underlie the summary data displayed in the following figure panels as they are essential for readers to assess your analysis and to reproduce it: Figs 1BCDE, 2B, 3ABCD, 4, S1, S2, S3, S4, S5, S6AB, S7, S8, S9, S10, S11, S12, S13, S14, S15, S16, S17, S18. NOTE: the numerical data provided should include all replicates AND the way in which the plotted mean and errors were derived (it should not present only the mean/average values).

We require the original, uncropped and minimally adjusted images supporting all blot and gel results reported in an article's figures or Supporting Information files. We will require these files before a manuscript can be accepted so please prepare and upload them now. Please carefully read our guidelines for how to prepare and upload this data: https://journals.plos.org/plosbiology/s/figures#loc-blot-and-gel-reporting-requirements

DATA NOT SHOWN?

REVIEWERS' COMMENTS:

Reviewer #1:

It is clear that the authors have made a substantial effort in addressing reviewer comments. The additional results and changes to the manuscript are sufficient to convince me that the the main conclusions of the manuscript are now supported by the data.

On remaining thing that left me confused is the "cumulative mutation rate" used in Fig A3 and Fig 1E. In reply R3.2 the mutation rate in Fig 1E is described as cumulative in a sense that in stationary phase there is no mutation rate so no new mutations are formed and the curve levels out. However in Fig 1E it is not explained that this is cumulative and the figure axis only says Mutation rate. In Fig A3 the axis legend says Cumulative mutation rate but for panel D and F the cumulative mutation rate goes to 0 after 24 h. Please clarify this to help the reader understand what is shown.

Reviewer #2:

No new comments.

Reviewer #3:

[identifies himself as Christopher J Marx]

I was generally quite positive about the R1 version I saw last, but I am even more convinced by this version. There were many excellent questions raised by the reviewers and editor and I feel like these have been dealt with in a very thorough manner. My concerns, which were mainly those of being able to "see" what the model is actually doing, are assuaged by the figures showing mutation rate as a function of time for all of the different models. I think that is extremely helpful to connect intuition to actual model outputs. I fully support this version of the paper.

---

## [Editor Report · Decision Letter 3]

13 Jun 2024

Dear Dr Green,

Thank you for the submission of your revised Update Article "Collective peroxide detoxification determines microbial mutation rate plasticity in E. coli" for publication in PLOS Biology. On behalf of my colleagues and the Academic Editor, Deepa Agashe, I'm pleased to say that we can in principle accept your manuscript for publication, provided you address any remaining formatting and reporting issues. These will be detailed in an email you should receive within 2-3 business days from our colleagues in the journal operations team; no action is required from you until then. Please note that we will not be able to formally accept your manuscript and schedule it for publication until you have completed any requested changes.

IMPORTANT: Many thanks for providing the email thread to support the authorship of Francesca Tyrrell. We now note the subsequent addition of James Letton, and I have asked my colleagues to include a request for equivalent support for his inclusion.

Sincerely, 

Roli Roberts

Senior Editor

PLOS Biology

rroberts@plos.org